# A standalone bismuth vanadate-silicon artificial leaf achieving 8.4% efficiency for hydrogen production

Boyan Liu[1], Xin Wang[1], Yingjuan Zhang[1], Mingshan Zhu [2] ✉, Chenxin Zhang[3], Shaobin Li [3], Yanhang Ma [4], Wei Huang [1] ✉ & Songcan Wang [1,5] ✉

The development of scalable photoelectrochemical water splitting with high solar-to-hydrogen efficiency and long-term stability is essential while challenging for practical application. Here, we design a $BiVO_4$ photoanode with gradient distributed oxygen vacancies, which induces strong dipole fields to promote charge separation. Growing sea-urchin-like FeOOH cocatalyst on the photoanode leads to a photocurrent density of 7.0 mA cm$^{-2}$ at 1.23 V versus the reversible hydrogen electrode and is stable for over 520 h under AM 1.5 G illumination. By integrating with a silicon photovoltaic cell, the standalone artificial leaf achieves a solar-to-hydrogen efficiency of 8.4%. The scale-up of these artificial leaves up to 441 cm$^2$ in size can deliver a solar-to-hydrogen efficiency of 2.7% under natural sunlight. Life cycle assessment analysis shows that solar water splitting has little environmental footprint for hydrogen production. Our study demonstrates the possibility of designing metal oxide-based artificial leaves for scalable solar hydrogen production.

The widespread utilization of hydrogen as a clean alternative to fossil fuels provides a feasible approach to address sustainability challenges due to the massive carbon emissions resulted from the increasing demand of energy. Currently, over 95% of global hydrogen production is derived from fossil fuels, which unfortunately contributes to non-negligible carbon emissions[1,2]. Therefore, it is highly desirable to develop a green hydrogen production technology to maintain the sustainable development of the human society[3,4]. The conversion of solar energy into hydrogen via a photoelectrochemical (PEC) water splitting process provides a promising approach for hydrogen production with zero emissions. Although worldwide efforts have been devoted for over half a century, high-performance and low-cost PEC water splitting devices with great long-term stability are still lacking[5,6]. Some high-performance PEC water splitting devices generally involve expensive III-V semiconductors as the photoelectrodes[7,8], which is

unfavorable for scale-up application. Recently, perovskite PEC cells have shown great potential for scalable solar water splitting. For example, a monolithic stacked silicon-perovskite tandem exhibited a solar-to-hydrogen (STH) efficiency of 20.8%[9]. However, the photo-current of the device degraded to 60% after 102 h of continuous operation. A Ni-encapsulated FAPbI$_3$ photoanode could stably split water for 72 h, and the assembled unbiased device with a size of 123 cm$^2$ achieved an STH efficiency of 8.5%[10]. How to develop an Earth-abundant photoelectrode with both high PEC activity and ultra-long stability is essential to drive the PEC water splitting technology towards practical application while challenging.

Owing to the relatively good stability in aqueous solutions under oxidative environment during PEC water splitting, metal oxide photoanodes have been intensively investigated in the past decades[11,12]. Bismuth vanadate ($BiVO_4$) has been recognized as one of the most

[1]State Key Laboratory of Flexible Electronics & Institute of Flexible Electronics, Northwestern Polytechnical University, 127 West Youyi Road, Xi'an 710072, China. [2]Guangdong Key Laboratory of Environmental Pollution and Health, School of Environment, Jinan University, Guangzhou 511443, China. [3]College of the Environment and Ecology, Xiamen University, Xiamen 361102, China. [4]School of Physical Science and Technology & Shanghai Key Laboratory of High-resolution Electron Microscopy, ShanghaiTech University, Shanghai 201210, China. [5]Research & Development Institute of Northwestern Polytechnical University in Shenzhen, Sanhang Science & Technology Building, No. 45th, Gaoxin South 9th Road, Nanshan District, Shenzhen 518063, China. ✉e-mail: zhumingshan@jnu.edu.cn; iamwhuang@nwpu.edu.cn; iamscwang@nwpu.edu.cn

promising oxide-based photoanodes for PEC water splitting due to the relatively narrow bandgap for visible light absorption[13]. In addition, the conduction band edge position of $BiVO_4$ is very close to the thermodynamic $H_2$ evolution potential, thus requiring a very low onset potential for PEC water oxidation[14]. However, the intrinsic drawbacks of low electron mobility ($0.044\ cm^2V^{-1}\ s^{-1}$) and short hole diffusion length (~70 nm) lead to severe charge recombination within the $BiVO_4$ film[15]. Consequently, the photocurrent densities of pristine $BiVO_4$ photoanodes for PEC water splitting are much lower than the theoretical value of $7.5\ mA\ cm^{-2}$ under AM 1.5 G illumination[16]. On the other hand, the long-term stability of pristine $BiVO_4$ during PEC water splitting is very poor because of the photoanodic corrosion issue[17]. Therefore, the design of active and stable $BiVO_4$ photoanode is essential to meet the demand for practical applications.

Here, we develop an electrolyte recipe for the preparation of $Bi_{34.7}O_{36}(SO_4)_{16}$ precursor films, followed by solid thermal reaction with vanadyl acetylacetonate ($VO(acac)_2$) to construct wormlike $BiVO_4$ photoanodes with a gradient distribution of oxygen vacancies (denoted as BVO-$\Delta O_v$). Systematical studies found that the BVO-$\Delta O_v$ photoanode can generate strong dipole fields within the film, thus significantly promotes bulk charge separation, achieving a photocurrent density of $7.2\ mA\ cm^{-2}$ at 1.23 V vs. the reversible hydrogen electrode (RHE) under AM 1.5 G illumination for sulfide oxidation. Upon the deposition of an ultrathin and pinhole-free FeOOH layer with a sea-urchin structure as the oxygen evolution cocatalyst (OEC), a photocurrent density of $7.0\ mA\ cm^{-2}$ at 1.23 V vs. RHE under AM 1.5 G illumination for PEC water splitting is achieved, which is the best performance among all reported $BiVO_4$-based photoanodes (Supplementary Table 1). Moreover, stable PEC water splitting can be performed for over 520 h without obvious fading. Upon integration with a silicon solar cell, the standalone artificial leaf with an exposed area of $0.126\ cm^2$ achieves an STH efficiency of 8.4% for water splitting under AM 1.5 G illumination. In addition, a scale-up artificial leaf with dimensions of 21 cm × 21 cm (exposed area: $306.25\ cm^2$) exhibits an STH efficiency of 2.7% under natural sunlight. Life cycle assessment (LCA) analysis shows that the PEC water splitting process has little environmental footprint compared to natural gas reforming and electrocatalytic water splitting for hydrogen production, demonstrating the great potential for scale-up solar hydrogen production in a sustainable manner.

## Results

### Synthesis and characterization of $BiVO_4$ photoanodes

To address the severe charge recombination issue in the bulk of $BiVO_4$ photoanodes while keeping the high light utilization efficiency, we first grew bismuth precursor films on fluorine-doped $SnO_2$ (FTO) glass substrates by an electrodeposition process (details are shown in the Method Section). The morphology of the bismuth precursor films can be tailored by tuning the applied potential between −0.3 and −0.7 V vs. the saturated Ag/AgCl electrode while fixing the electrodeposition time at 2 min. When the applied potential is −0.3 V, the obtained bismuth precursor film is composed of cauliflower-like particles with a diameter of about 2.1 μm scattering on the FTO substrate, and gaps between the particles can be clearly observed (Fig. 1a). Interestingly, when the applied potential is reduced to −0.5 V, homogeneous sponge-like structures with enriched nanopores can be observed (Fig. 1b). However, when the applied potential is further reduced to −0.7 V, the film is composed of plate-like particles connecting together (Fig. 1c). Crystalline structures of the obtained bismuth precursor films were characterized by X-ray diffraction (XRD). As shown in Fig. 1d, along with the strong FTO signals, all peaks can be assigned to $Bi_{34.7}O_{36}(SO_4)_{16}$ (JCPDS No. 73-6692), so the bismuth precursor films prepared at different applied potentials of −0.3, −0.5, and −0.7 V are denoted as BSO-3, BSO-5, and BSO-7, respectively.

After thermal reacting the as-prepared BSO-$x$ precursor films with vanadyl acetylacetonate ($VO(C_5H_7O_2)_2$) and the removal of excessive $V_2O_5$, wormlike $BiVO_4$ films were obtained and the corresponding samples were denoted as BVO-$\Delta O_v$-$x$ ($x$ = 3, 5 and 7). As shown in Fig. 1e–g, the BVO-$\Delta O_v$-3, BVO-$\Delta O_v$-5, and BVO-$\Delta O_v$-7 samples converted from the BSO-$x$ precursor films prepared at different applied potentials exhibit very similar wormlike morphologies. The average size of the wormlike particles is approximately 400 nm. Aggregation of the particles can be observed in the BVO-$\Delta O_v$-3 film (Fig. 1e), which is consistent with its BSO-3 precursor film (Fig. 1a). The film thickness is approximately 2.3 μm. However, the BVO-$\Delta O_v$-5 and BVO-$\Delta O_v$-7 films are composed of homogeneously distributed wormlike particles with a film thickness of 1.2 and 1.7 μm, respectively (Fig. 1f, g). According to their XRD patterns (Fig. 1h), all XRD peaks can be indexed to monoclinic $BiVO_4$ (JCPDS No. 14-0688) along with the FTO signals[18,19]. Moreover, all samples show similar peak shapes and intensities, suggesting that different deposition potentials do not affect the crystal facet orientation of $BiVO_4$.

Figure 1i demonstrates a transmission electron microscopy (TEM) image of BVO-$\Delta O_v$-5, where a particle with a size of around 400 nm can be observed, which is consistent with the SEM image. In addition, high-resolution TEM (HRTEM) image of BVO-$\Delta O_v$-5 (Fig. 1j) shows clear arrangement of the atoms, which matches well to the monoclinic $BiVO_4$ (JCPDS No. 14-0688). Moreover, the selected area electron diffraction (SAED) pattern of BVO-$\Delta O_v$-5 exhibits clear matrix spots (Fig. 1k), suggesting the single crystalline feature[20]. The uniform distribution of Bi, V and O in the BVO-$\Delta O_v$-5 particles is revealed by EDS mapping analysis (Fig. 1l).

### Optoelectrical properties of $BiVO_4$ photoanodes

Bulk charge separation efficiencies of the BVO-$\Delta O_v$-$x$ photoanodes were measured in a three-electrode cell with 0.2 M $Na_2SO_3$ as the hole scavenger under AM 1.5 G illumination ($100\ mW\ cm^{-2}$). The Xe light equipped with an AM 1.5 G filter was carefully calibrated to well match the standard AM 1.5 G spectrum in the range of 300–800 nm (Supplementary Fig. 1). Owing to the low activation energy and fast kinetics for the oxidation of $SO_3^{2-}$ ions[21], all photogenerated holes reaching the surface of BVO-$\Delta O_v$-$x$ would be immediately consumed in the presence of $SO_3^{2-}$ ions, and thus the effect of surface charge recombination on the photocurrent densities can be excluded. As shown in Supplementary Fig. 2a, the BVO-$\Delta O_v$-5 photoanode exhibits a photocurrent density of $7.2\ mA\ cm^{-2}$ at 1.23 V vs. RHE under AM 1.5 G illumination, which is 96% of its theoretical maximum. In contrast, the BVO-$\Delta O_v$-3 and BVO-$\Delta O_v$-7 photoanodes exhibit a photocurrent density of 4.0 and $5.5\ mA\ cm^{-2}$ at 1.23 V vs. RHE, respectively. To obtain more reliable results, the photocurrent densities of 16 pieces of each BVO-$\Delta O_v$-$x$ photoanode at 1.23 V vs. RHE under AM 1.5 G illumination were collected. Supplementary Fig. 2b shows the corresponding histograms of the BVO-$\Delta O_v$-3, BVO-$\Delta O_v$-5 and BVO-$\Delta O_v$-7 photoanodes. It can be observed that the average photocurrent density of the BVO-$\Delta O_v$-5 films is $6.5\ mA\ cm^{-2}$ with a champion value of $7.2\ mA\ cm^{-2}$, while those of the BVO-$\Delta O_v$-3 and BVO-$\Delta O_v$-7 samples are 3.02 and $5.24\ mA\ cm^{-2}$, respectively. Hereafter, all discussion on the BVO-$\Delta O_v$ films refers to the BVO-$\Delta O_v$-5 films, unless stated otherwise.

It should be mentioned that the electrodeposition time also affects the morphology and thickness of the obtained BSO precursor films and the converted BVO-$\Delta O_v$ films (Supplementary Discussion and Supplementary Fig. 3a–f). When keeping the electrodeposition potential at −0.5 V while changing the electrodeposition time to 1, 3, and 4 min, the PEC sulfide oxidation performances of these samples are different but lower than that of their BVO-$\Delta O_v$ counterpart (Supplementary Fig. 4). Moreover, the recipe in the electrolyte for fabricating the bismuth precursor films also affects the PEC performance of the converted $BiVO_4$ films (Supplementary Figs. 5, 6). Interestingly,

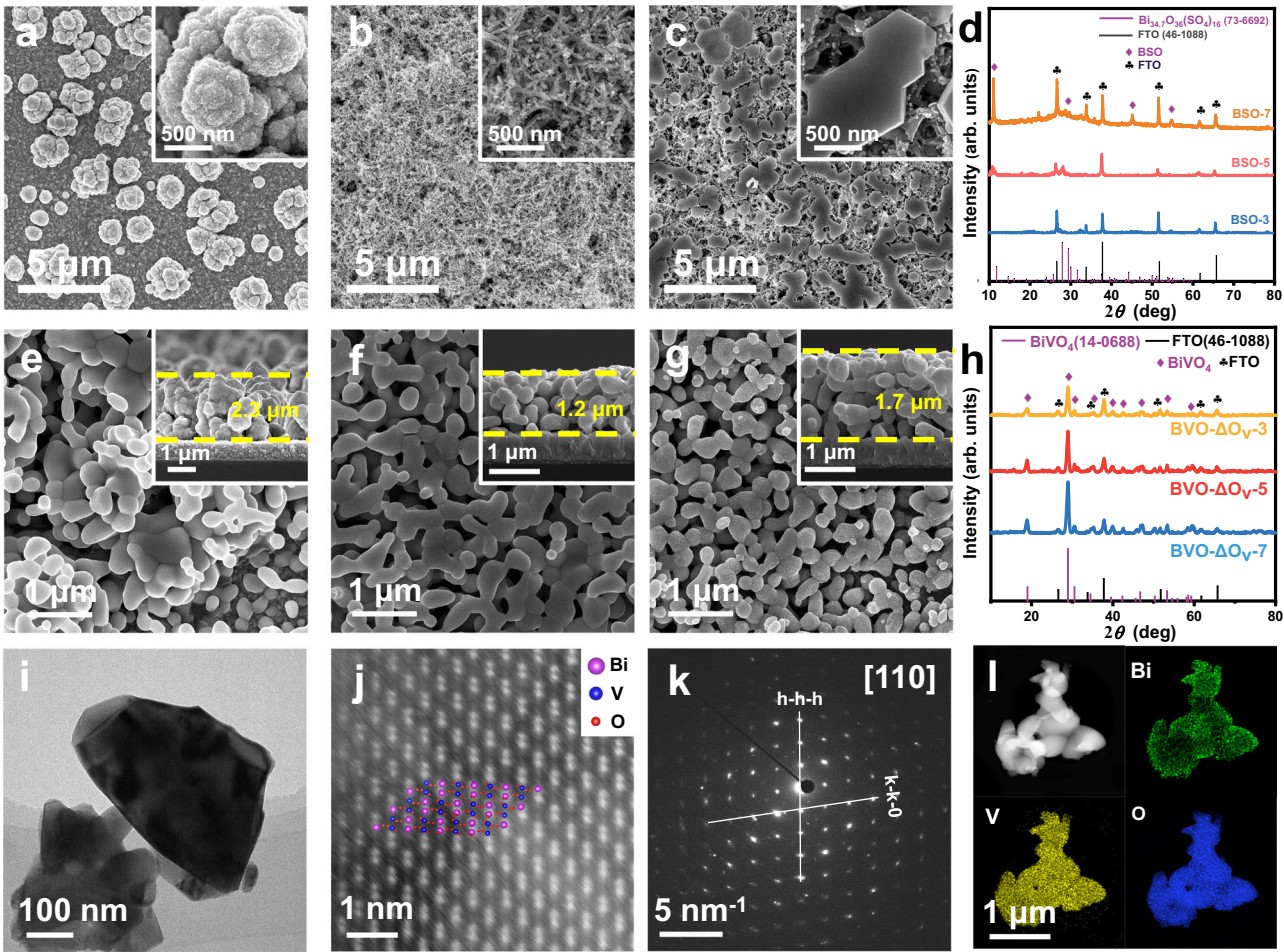

**Fig. 1 | Material characterizations of the samples.** SEM images of (**a**) BSO-3, (**b**) BSO-5 and (**c**) BSO-7. Insets: local magnification of the BSO-*x* samples. **d** XRD patterns of BSO-3, BSO-5 and BSO-7. SEM images of (**e**) BVO-$\Delta O_v$-3, (**f**) BVO-$\Delta O_v$-5 and (**g**) BVO-$\Delta O_v$-7. Insets: the cross-sectional SEM images of the BVO-$\Delta O_v$-*x* samples.

**h** XRD patterns of BVO-3, BVO-5 and BVO-7. **i** TEM image, (**j**) HRTEM image, (**k**) SAED pattern, and (**l**) Element mappings of BVO-$\Delta O_v$-5. Source data are provided as a Source Data file.

when the nitric acid is replaced by acetic acid with the same pH value in the electrolyte, the bismuth precursor film can be obtained under the same electrodeposition conditions. Although the converted $BiVO_4$ film (denoted as BVO) exhibits a similar wormlike structure and film thickness (Supplementary Fig. 5c), the PEC performance is much lower than its BVO-$\Delta O_v$ counterpart. To understand the underlying mechanisms, the optoelectrical performance of the BVO-$\Delta O_v$ and BVO samples were further studied.

Figure 2a shows the photocurrent density versus potential curves of BVO-$\Delta O_v$ and BVO. Although their onset potentials are very similar, BVO-$\Delta O_v$ exhibits much better photocurrent densities in the whole applied potential range. According to the UV-vis light absorption curves, the bandgaps of BVO-$\Delta O_v$ and BVO are 2.40 and 2.43 eV, respectively (Supplementary Fig. 7). Incident-photon-to-current conversion efficiency (IPCE) curves collected in a 1 M borate buffer electrolyte with 0.2 M $Na_2SO_3$ at 1.23 V vs. RHE under AM 1.5 G illumination illustrate that BVO-$\Delta O_v$ exhibits an IPCE value of 98% in the wavelength range of 300-400 nm, which is almost double that of its BVO counterpart (Fig. 2b). Since $Na_2SO_3$ can be oxidized by the photogenerated holes with almost no energy loss[22], the photocurrent densities and IPCE values reflect the charge separation properties of the samples. By integrating the IPCE curves of BVO-$\Delta O_v$ and BVO with the standard AM 1.5 G spectrum, the estimated photocurrent densities are 6.9 and 3.8 mA cm$^{-2}$, respectively (Supplementary Fig. 8), which are close to the measured photocurrent densities shown in Fig. 2a. In the absence of $Na_2SO_3$, the photocurrent densities and IPCE values of both BVO-

$\Delta O_v$ and BVO decrease (Supplementary Discussion and Supplementary Fig. 9) due to the sluggish kinetics for OER.

According to Supplementary Equation (8), the charge separation efficiency ($\eta_{sep}$) of BVO-$\Delta O_v$ is nearly 100% at 1.23 V vs. RHE, while its BVO counterpart only exhibits a $\eta_{sep}$ of 50% at the same applied potential (Fig. 2c). Electrochemical impedance spectroscopy (EIS) curves demonstrate that the charge transfer resistance of BVO-$\Delta O_v$ is significantly lower than that of its BVO counterpart (Supplementary Discussion, Supplementary Fig. 10 and Supplementary Table 2), suggesting the better surface charge transfer properties that alleviate the energy loss during PEC water splitting[23].

Mott-Schottky (MS) curves of BVO-$\Delta O_v$ and BVO films were measured in the dark to further understand the charge transport properties of the samples. As shown in Fig. 2d, both photoanodes show a positive slope in the MS curves, showing n-type characteristics. In addition, both photoanodes have similar flat-band potentials, but the slope of BVO-$\Delta O_v$ is flatter, indicating that the carrier density is higher, which is beneficial for bulk charge transport and separation[24]. The carrier density is calculated according to Supplementary Equation (5). As shown in Supplementary Table 3, the carrier density of BVO-$\Delta O_v$ almost doubles that of its BVO counterpart, which is beneficial for the transport of photogenerated charge carriers within the film.

To further understand the underlying mechanism of the significantly enhanced charge separation efficiency, Kelvin probe force microscopy (KPFM) was applied to detect the surface potential profiles of BVO-$\Delta O_v$ and BVO in the dark and under a Xe lamp illumination. As

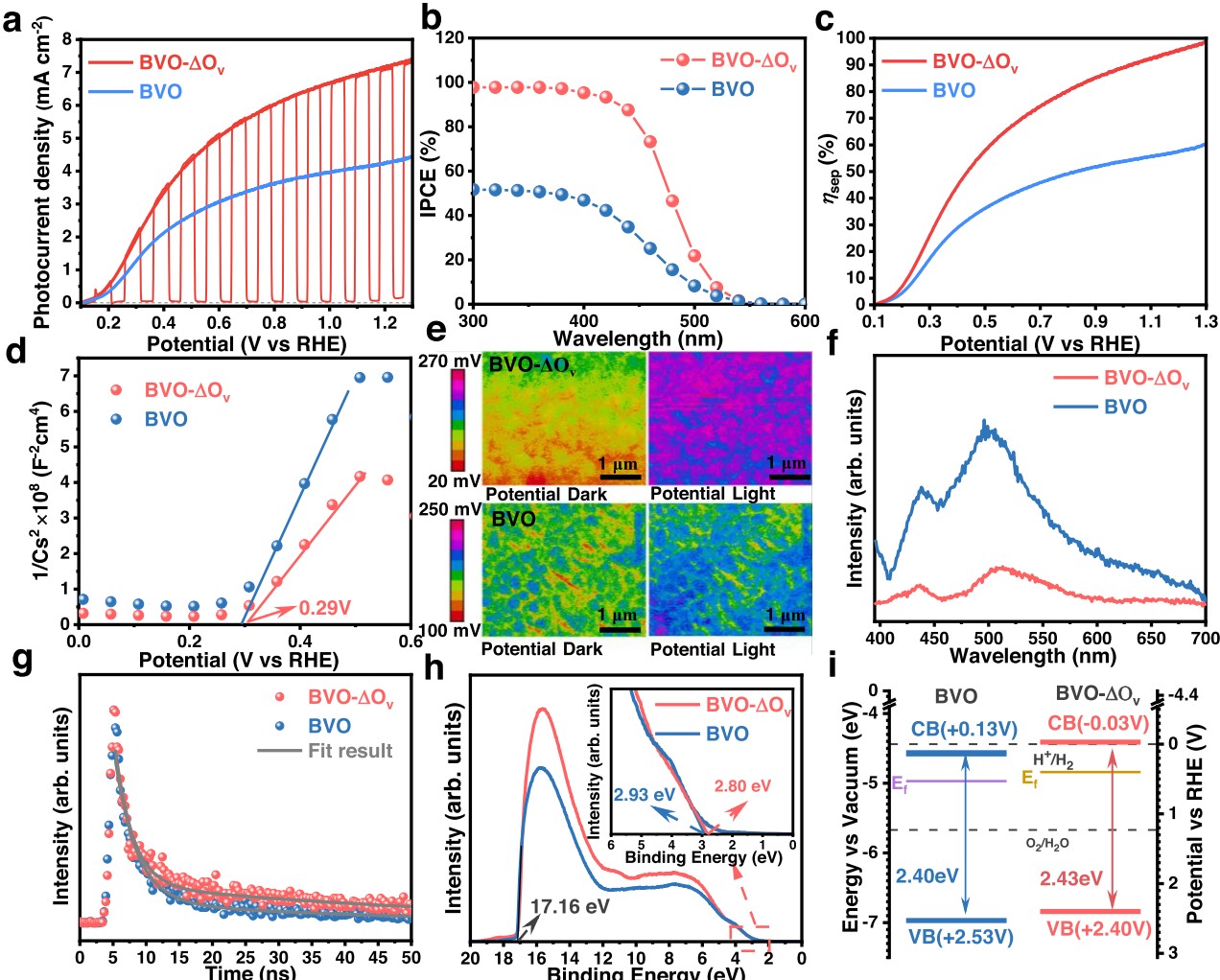

**Fig. 2 | Photoelectrochemical properties of the samples. a** Photocurrent density versus potential curves of BVO-ΔOᵥ and BVO in a 1 M borate buffer electrolyte with 0.1 M Na₂SO₃ (pH 9.5) under AM 1.5 G illumination. **b** IPCE curves at 1.23 V vs. RHE, (**c**) $\eta_{sep}$ curves, (**d**) MS curves under dark, (**e**) KPFM images, (**f**) PL spectra, (**g**) Time-resolved fluorescence emission spectra, (**h**) UPS curves, inset: local magnification, (**i**) Energy level diagram of the BVO-ΔOᵥ and BVO films. Source data are provided as a Source Data file.

shown in Fig. 2e, the KPFM images of BVO-ΔOᵥ exhibit significantly different surface potentials under dark and light conditions, suggesting the generation of a high photovoltage as the driving force for charge separation[25]. However, the KPFM images of BVO under dark and light conditions only have a slight difference, indicating the generation of a relatively small photovoltage. As listed in Supplementary Table 4, the $\Delta P_{avg}$ of BVO-ΔOᵥ is as high as 101.3 mV, which is around 7 times higher than that of its BVO counterpart. Photoluminescence (PL) spectroscopy demonstrates that BVO-ΔOᵥ exhibits a much weaker band-to-band emission peak at around 510 nm compared to its BVO counterpart (Fig. 2f), indicating the higher charge separation efficiency[26]. The carrier lifetime of BVO-ΔOᵥ and BVO were further investigated by nanosecond time-resolved fluorescence (TR-PL, Fig. 2g), and their average carrier lifetime ($\tau_{avg}$) values were calculated according to Supplementary Equation (13). As listed in Supplementary Table 5, the average carrier lifetime of BVO-ΔOᵥ is 42.40 ns, which is around 1.86 times higher than its BVO counterpart (22.83 ns).

To determine the band structures of BVO-ΔOᵥ and BVO, ultraviolet photoelectron spectroscopy (UPS) was used to investigate the Fermi level ($E_f$) of the semiconductors and the valence band maximum with respect to the $E_f$. The relationship between these energy levels in the UPS spectra of BVO-ΔOᵥ and BVO is shown in Fig. 2h. The values of $E_f$ for BVO-ΔOᵥ and BVO are 4.84

and 4.97 eV versus vacuum energy ($E_{vac}$), respectively, and the valence band positions are 6.84 and 6.97 eV versus $E_{vac}$ (corresponding to 2.40 and 2.53 eV versus RHE), respectively. Based on their bandgaps, the conduction band minimum (CBM) values of BVO-ΔOᵥ and BVO are −0.03 eV and +0.13 eV vs. RHE, respectively. Thus, the band structures of BVO-ΔOᵥ and BVO are demonstrated in Fig. 2i. It can be observed that the CBM of BVO-ΔOᵥ is slightly more negative than that of BVO, which is beneficial for PEC water splitting with a low onset potential.

## Characterization of oxygen vacancy distribution in BiVO₄ photoanodes

To deeply unveil the underlying mechanism for the significantly different optoelectrical properties of BVO-ΔOᵥ and BVO, electron paramagnetic resonance (EPR) was carried out to investigate their crystalline structures in detail. As shown in Fig. 3a, a signal centered at $g = 2.00173$ is observed for both samples, indicating the existence of oxygen vacancies[27]. The signal for BVO-ΔOᵥ is much stronger than that of its BVO counterpart, suggesting the formation of more oxygen vacancies in the bulk. In addition, Raman spectra demonstrate that the V−O bond length of BVO-ΔOᵥ is shorter than BVO (Supplementary Discussion and Supplementary Fig. 11), which also support the formation of more oxygen vacancies[28].

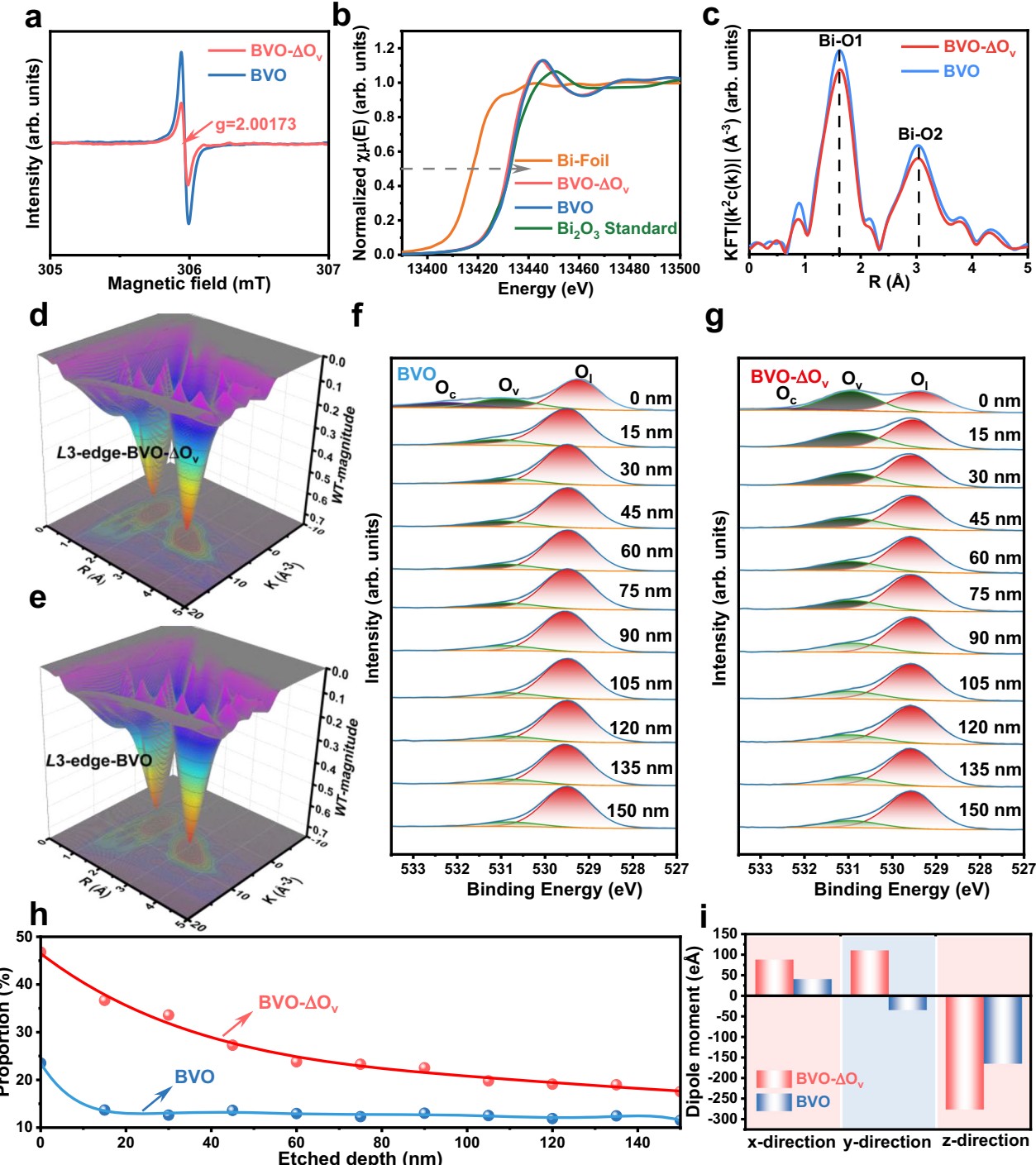

**Fig. 3 | Characterizations of oxygen vacancies and the induced dipole moment.** **a** Electron spin resonance spectra, (**b**) Normalized Bi $L3$-edge XANES $\mu(E)$ spectra, (**c**) Bi $L3$-edge radial distance $\chi(R)$ space spectra, and (**d**–**e**) Bi $L3$-edge 3D contour wavelet transforms of BVO-$\Delta O_v$ and BVO. Etching-XPS spectra of O $1s$ orbits for (**f**) BVO and (**g**) BVO-$\Delta O_v$ with the etching depth from 0 to 150 nm. **h** Concentration profile of $O_v$ species as a function of etching depth for BVO-$\Delta O_v$ and BVO. **i** The calculated dipole moments of BVO-$\Delta O_v$ and BVO along three different crystallographic directions. Source data are provided as a Source Data file.

The Bi $L3$-edge X-ray absorption near edge structure (XANES) measurements of BVO-$\Delta O_v$ and BVO indicate that the oxidation state is mainly $Bi^{3+}$ through comparing with the reference materials of metal Bi foil and $Bi_2O_3$ (Fig. 3b). In addition, the radial distance space spectra $\chi(R)$ of Bi in BVO-$\Delta O_v$ and BVO, and their corresponding references were analyzed (Supplementary Figs. 12–14), which provides more convincing support for the generation of oxygen vacancies. As shown in Fig. 3c, the peaks located at approximately 1.63 Å assigned to the Bi–O1 bond are consistently observed in both the Bi $L3$-edge curves of the BVO-$\Delta O_v$ and BVO samples. In addition, the peaks located at approximately 3.02 Å assigned to the Bi–O2 bond can also be observed in the BVO-$\Delta O_v$ and BVO samples. Noticeably, the peaks of Bi–O1 and Bi–O2 bonds in BVO-$\Delta O_v$ are weaker than those of BVO, indicating the generation of more oxygen vacancies that results in less amount of Bi–O1 and Bi–O2 bonds[29]. No noticeable differences can be observed in the Bi $L3$-edge 3D contour wavelet transforms of the samples (Fig. 3d, e).

The formation of oxygen vacancies in the BVO-$\Delta O_v$ sample can be further verified by the results of quantitative $\chi(R)$ space spectra fitting and wavelet transform of $\chi(k)$. As shown in Supplementary Table 6, Bi−O bond with little different coordination numbers can be confirmed. The good fitting results of $\chi(R)$ and $\chi(k)$ space spectra (Supplementary Fig. 13) with reasonable $R$-factors and the obtained fitting parameters (Supplementary Table 6) provide a quantitative of the Bi−O1 and Bi−O2 bonds. The shorter distances of the Bi−O bonds in BVO-$\Delta O_v$ suggest the presence of more oxygen vacancies, which is consistent with their EPR results. XANES spectra at V $K$-edge of BVO-$\Delta O_v$ and BVO also confirm the presence of more oxygen vacancies in BVO-$\Delta O_v$ (Supplementary Discussion, Supplementary Fig. 15, and Supplementary Table 7)[30].

To understand oxygen vacancy distribution in BVO-$\Delta O_v$ and BVO, argon-ion etching-XPS was conducted to collect the depth profiles of O $1s$ in the BVO-$\Delta O_v$ and BVO samples. The spectra were collected at each etching step length of 15 nm (a total of 150 nm from the outermost surface inward). As shown in Supplementary Fig. 16, the spectra of different etching depths contain similar B, V and O signals, while the intensity of the surface adsorbed contamination C $1s$ peak decreases gradually with the increase of etching depth. The binding energy of Bi $4f$ and V $2p$ does not change significantly before and after etching (Supplementary Figs. 17, 18). Interestingly, with the increase of etching depth, low-valent Bi ion peaks appear and increase gradually, possibly caused by the reduction of $Bi^{3+}$ during argon ion etching[31]. Interestingly, only the surface of BVO contains oxygen vacancies, while the oxygen vacancies significantly decreased in the bulk and no obvious change is observed with the increase of depth (Fig. 3f). However, the surface of BVO-$\Delta O_v$ contains much more oxygen vacancies (Fig. 3g). With the increase of depth, the oxygen vacancies gradually decrease, demonstrating an interesting gradient distribution within the film (Fig. 3h).

To deeply understand the effect of gradient-distributed oxygen vacancies on the charge separation properties, a $BiVO_4$ model with gradient-distributed oxygen vacancies and the other with uniform distributed oxygen vacancies were constructed (Supplementary Fig. 19), and their dipole moments were calculated. As shown in Fig. 3i, BVO exhibits dipole moments of 50, −50, and −150 e$\text{Å}$ along the $x$-, $y$-, and $z$-directions, respectively. However, the dipole moment along the $x$- direction is significantly enhanced to 90 e$\text{Å}$ in BVO-$\Delta O_v$. In particular, much greater enhancements are observed in the $y$- and $z$-directions, with the values of 125 and −270 e$\text{Å}$ along the $y$- and $z$- directions in BVO-$\Delta O_v$. Therefore, the gradient-distributed oxygen vacancies in BVO-$\Delta O_v$ induce strong dipole fields along the (100), (010) and (001) directions, providing extra driving forces for the separation and transfer of photogenerated electron-hole pairs, which alleviates bulk charge recombination in the photoanode. The photocurrent density of BVO-$\Delta O_v$ drops significantly when the oxygen vacancies are removed (Supplementary Discussion and Supplementary Fig. 20), suggesting the critical role of gradient-distributed oxygen vacancies in enhancing charge separation in the bulk of the photoanode.

## PEC water splitting performance of BVO-$\Delta O_v$ with an external bias

Loading the photoanode surfaces with a proper OEC is essential for PEC water splitting. The BVO-$\Delta O_v$ samples were decorated by FeOOH, NiOOH, NiFeOOH and NiFeCoOOH OECs using a photo-assisted electrodeposition process, and the obtained samples were denoted as BVO-$\Delta O_v$/FeOOH, BVO-$\Delta O_v$/NiOOH, BVO-$\Delta O_v$/NiFeOOH, and BVO-$\Delta O_v$/NiFeCoOOH, respectively. Since phosphate buffer electrolytes can slowly dissolve $BiVO_4$ and Ni-based catalysts[17,32], PEC water splitting performance of all samples were measured in a 1 M borate buffer without a sacrificial agent. Although BVO-$\Delta O_v$/NiFeCoOOH exhibits a slightly higher photocurrent density than BVO-$\Delta O_v$/FeOOH at 1.23 V vs. RHE, the stability is poor (Supplementary Discussion and

Supplementary Fig. 21). Therefore, FeOOH was selected as the OEC for further investigation. As shown in Fig. 4a, the BVO-$\Delta O_v$ particle surfaces are fully coated by FeOOH needles, forming sea urchin structures. The enlarged SEM image (Fig. 4b) demonstrates that no pinholes can be observed. EDS mapping confirms the homogeneous distribution of Bi, V, O, and Fe (Supplementary Fig. 22), suggesting the successful coating of FeOOH OECs. Moreover, the HRTEM image illustrates that the thickness of FeOOH is around 10 nm and the clear fringes of FeOOH can be observed, indicating the high crystallinity of the FeOOH OEC layer (Supplementary Fig. 23).

As shown in Fig. 4c, the BVO-$\Delta O_v$/FeOOH photoanode exhibits a lower overpotential and steeper water oxidation current density than the pristine BVO-$\Delta O_v$ photoanode, indicating the excellent electrocatalytic activity of FeOOH for OER. To calculate the surface charge separation efficiencies, the photocurrent density versus potential curves of BVO-$\Delta O_v$ and BVO-$\Delta O_v$/FeOOH in a 1 M borate buffer electrolyte with and without 0.1 M $Na_2SO_3$ (pH 9.5) under AM 1.5 G illumination were collected (Supplementary Fig. 24). According to Supplementary Equation (9), the BVO-$\Delta O_v$/FeOOH photoanode exhibits a surface charge transfer efficiency of 94.9% (Fig. 4d), which is much higher than that of pristine BVO-$\Delta O_v$ (51.6%). The photocurrent density of BVO-$\Delta O_v$ is 3.74 mA cm$^{-2}$ at 1.23 V vs. RHE, whereas the BVO-$\Delta O_v$/FeOOH photoanode exhibits a photocurrent density of 7.0 mA cm$^{-2}$ (Fig. 4e), which is among the best performance reported for $BiVO_4$-based photoanodes, as summarized in Supplementary Table 1. The ABPE values derived from the $J$-$V$ plots were also calculated (Supplementary Equation (2)). As demonstrated in Fig. 4f, the BVO-$\Delta O_v$/FeOOH photoanode reveals a high ABPE value of 2.78% at a low applied potential of 0.62 V vs. RHE, which is the highest amongst all reported $BiVO_4$-based photoanodes (Supplementary Table 1). The onset potential ($V_{on}$) is determined by the intersection point of the $J$-$V$ plot subtracting the contribution of the dark current curve. The onset potential of the BVO-$\Delta O_v$/FeOOH photoanode is as low as 0.23 V, demonstrating that this photoanode is promising to be coupled with another photocathode or photovoltaic device for unbiased PEC water splitting. For possible scale-up applications, the geometrical area effect of the BVO-$\Delta O_v$/FeOOH photoanode on the photocurrent density was systematically studied. As demonstrated in Supplementary Fig. 25, the photocurrent density decreases with the increased area of the BVO-$\Delta O_v$/FeOOH photoanode.

Since the BVO-$\Delta O_v$/FeOOH photoanode exhibits the highest ABPE value at around 0.6 V vs. RHE (Fig. 4f) and the photoanode is generally operated at around 0.6 V vs. RHE in a photoanode-photocathode tandem device, we measured the long-term stability performance of a BVO-$\Delta O_v$/FeOOH film at 0.6 V vs. RHE. We found that the fluctuation of the temperature in the electrolyte affects the stability curve (Supplementary Fig. 26). To avoid the effect of temperature on the stability measurement, we designed a reactor with constant temperature circulating water to keep the temperature of the electrolyte at 25 °C during the stability test. As shown in Fig. 4g, the BVO-$\Delta O_v$/FeOOH film exhibits a very stable photocurrent density of ≈4.55 mA cm$^{-2}$ at 0.6 V vs. RHE under consecutive AM 1.5 G illumination for 500 h, exhibiting great potential for practical applications. To investigate the underlying mechanisms for long-term stability, the BVO-$\Delta O_v$/FeOOH film after 500 h overall water splitting measurement was characterized by SEM and XRD. It can be observed that the $BiVO_4$ particles are still fully covered by the FeOOH needle-like particles (Supplementary Fig. 27) and the XRD pattern of the BVO-$\Delta O_v$/FeOOH film after long-term stability measurement is almost the same as that of its fresh counterpart (Supplementary Fig. 28), indicating the excellent stability. During the stability test, slight fluctuations of the photocurrent densities can still be observed, which is due to the fluctuation of the output power of the Xe lamp during the long-term operation.

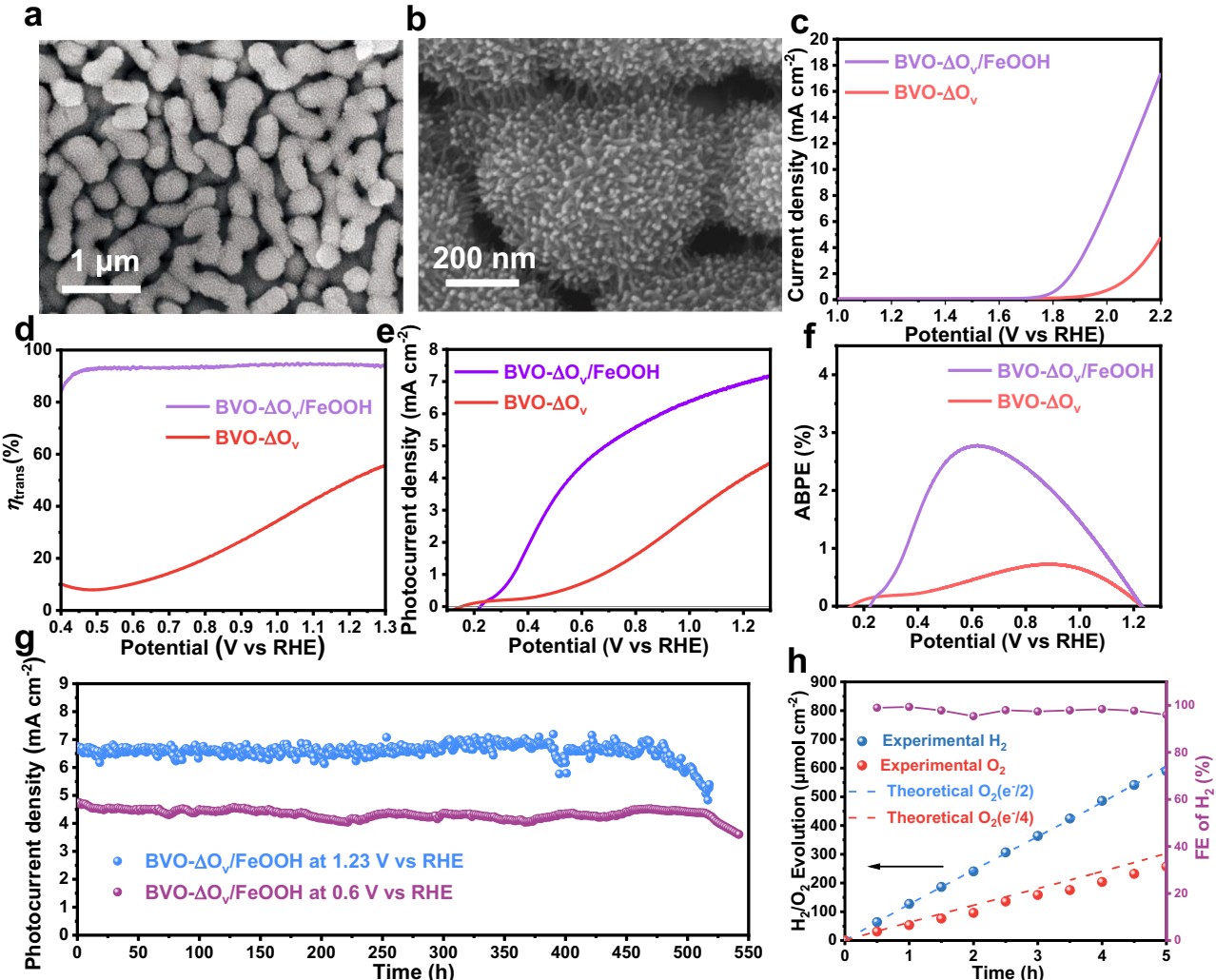

**Fig. 4 | Material characterizations and photoelectrochemical performances.** **a** SEM image of BVO-$\Delta O_v$/FeOOH. **b** Enlarged SEM image of BVO-$\Delta O_v$/FeOOH. **c** LSV curves in the dark, (**d**) Charge transfer efficiency ($\eta_{trans}$) curves, (**e**) LSV curves in the light, (**f**) ABPE curves of the BVO-$\Delta O_v$ and BVO-$\Delta O_v$/FeOOH photoanodes in a 1 M borate buffer electrolyte (pH 9.5) under AM 1.5 G illumination. **g** $J-t$ curve of the BVO-$\Delta O_v$/FeOOH photoanode at 0.6 and 1.23 V vs. RHE under AM 1.5 G illumination in a 1 M borate buffer electrolyte (pH 9.5). **h** Plots of the theoretical charge number obtained from the $J-t$ curves collected at 1.23 V vs. RHE and actual quantities of $H_2$ and $O_2$ evolution of a BVO-$\Delta O_v$/FeOOH photoanode. Exposed area of the photoanode: 0.126 cm². Source data are provided as a Source Data file.

The gradual decay of the photocurrent density can be observed after around 520 h, and the photocurrent density is decreased to 3.6 mA cm⁻² at 540 h. According to the SEM image (Supplementary Fig. 29), some FeOOH particles disappear which leads to the direct exposure of the $BiVO_4$ particles to the electrolyte. Therefore, photocorrosion of $BiVO_4$ leads to the gradual decay of the photocurrent density. The BVO-$\Delta O_v$/FeOOH film also shows excellent stability at 1.23 V vs. RHE for 470 h, and gradual decay of the photocurrent density can be observed in the range of 470–520 h (Fig. 4g and Supplementary Discussion). Similarly, the decay of photocurrent density is also attributed to the dissolution of FeOOH particles that loses the protection of $BVO_4$ particles (Supplementary Fig. 30). To evaluate the water splitting performance of the BVO-$\Delta O_v$/FeOOH in a sealed cell, $J-t$ curve was performed at 1.23 V vs. RHE under AM 1.5 G illumination in a 1 M borate buffer electrolyte for 5 h, and the produced gases were detected by a gas chromatography (GC) every 0.5 h. As shown in Fig. 4h, the produced $H_2$ and $O_2$ gases are 589.7 and 257.4 μmol cm⁻² after 5 h, respectively, indicating a stoichiometric ratio of around 2:1 for water splitting with an average faradaic efficiency of 97.6% (Supplementary Equation 15).

## Artificial leaves for unbiased solar water splitting

The favorable onset potential, along with its high performance and stability, suggest the promising potential of this photoanode for unbiased PEC water splitting when coupled with either a photocathode or a photovoltaic device. In this study, standalone artificial leaves with dimensions of 1 cm × 1 cm, 3 cm × 3 cm, 6 cm × 6 cm, 9 cm × 9 cm, 12 cm × 12 cm, and 21 cm × 21 cm were fabricated by integrating with a silicon cell (Supplementary Figs. 31–34). The J-V curves of a Si PV with and without the surface covered by a BVO-$\Delta O_v$/FeOOH photoanode are shown in Supplementary Fig. 35. Detailed information of the $J_{sc}$, $V_{oc}$, FF, and PCE of the Si PV panel is summarized in Supplementary Table 8. The circuit connection mechanism and the charge transfer properties of the artificial leaf are shown in Fig. 5a, b. Detailed connections between the BVO-$\Delta O_v$/FeOOH photoanodes and the Si PV panel are shown in Supplementary Figs. 32, 33. The artificial leaf with dimensions of 3 cm × 3 cm can continuously generate $H_2$ and $O_2$ bubbles under Xe lamp light (AM 1.5 G, 100 mW cm⁻², Supplementary Fig. 36a and Supplementary Movie 1). A large area artificial leaf with dimensions of 21 cm × 21 cm can generate observable $H_2$ and $O_2$ bubbles under natural sunlight (Fig. 5c, Supplementary Fig. 36b, and Supplementary Movie 2). An operating photocurrent density of

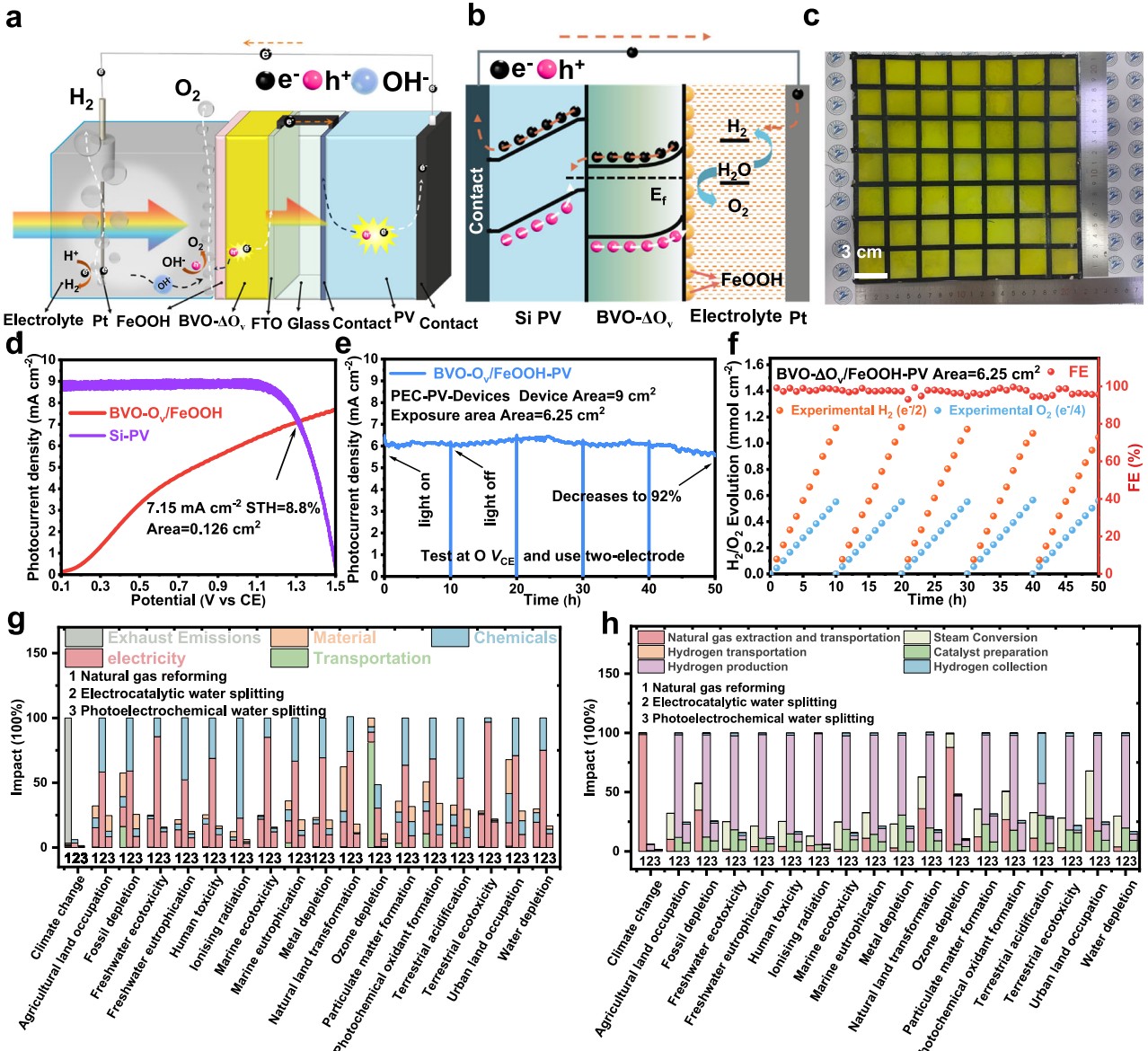

**Fig. 5 | Unassisted solar water splitting with a BiVO₄-PV artificial leaf.** Scheme of a BVO-ΔOᵥ/FeOOH-PV artificial leaf with circuit connection (**a**) and charge transportation (**b**). **c** Digital image of a wireless BVO-ΔOᵥ/FeOOH-PV artificial leaf with dimensions of 21 cm × 21 cm (exposed area: 306.25 cm²). **d** J–V curves of a BVO-ΔOᵥ/FeOOH photoanode and PV behind the BVO-ΔOᵥ/FeOOH photoanode in tandem with an exposed area of 0.126 cm² under AM 1.5 G irradiation (100 mW cm⁻²). **e** Photocurrent density-time curve of a wired BVO-ΔOᵥ/FeOOH-PV artificial leaf with

dimensions of 3 cm × 3 cm (exposed area: 6.25 cm²) under AM 1.5 G irradiation. **f** The corresponding H₂ and O₂ evolution of a of a wired BVO-ΔOᵥ/FeOOH-PV artificial leaf with dimensions of 3 cm × 3 cm (exposed area: 6.25 cm²) under AM 1.5 G irradiation. Relative environmental impacts of the three scenarios for hydrogen production (normalized to the highest value among three scenarios for each impact category): **g** classified by resources and emissions, and (**h**) classified by unit process. Source data are provided as a Source Data file.

7.15 mA cm⁻² corresponding to an unbiased STH efficiency of 8.8% (Supplementary Equation 18) is achieved for the artificial leaf with an exposed area of 0.126 cm² (Fig. 5d). Based on the operating photocurrent densities, the STH efficiencies of other artificial leaves with different exposed area of 1, 6.25, 25, 56.25, 100, and 306.25 cm² are 8.6%, 7.9%, 7.1%, 6.1%, 5.0%, and 3.5%, respectively (Supplementary Fig. 37).

A wired BVO-ΔOᵥ/FeOOH-PV artificial leaf with dimensions of 3 cm × 3 cm was placed in a sealed reactor connecting to an electrochemical workstation to monitor the photocurrent densities (Supplementary Fig. 38). The BVO-ΔOᵥ/FeOOH-PV artificial leaf can achieve water splitting for 50 h with a photocurrent density retention rate of 92% (Fig. 5e). It should be mentioned that the encapsulation in the circuit of the artificial leaf to avoid the direct contact of the electrolyte is very important to achieve long-term stability. We found

that if the encapsulation is not well, the photoelectrochemical corrosion of the circuit causes the fluctuation of the output photocurrent densities (Supplementary Fig. 39). Furthermore, we conducted long-term water splitting performance tests in a 1 M borate buffer electrolyte under AM 1.5 G illumination, and the gas production was detected via GC every hour. The system was vacuumed every 10 h to remove all gases for one cycle test. As shown in Fig. 5f, the generated H₂/O₂ show no noticeable decrease, with production rates of 1117/551 μmol cm⁻² in 10 h of consecutive illumination, with a ratio of round 2:1. This confirms the long-term operational capability of the fabricated artificial leaf. Since the BVO-ΔOᵥ/FeOOH photoanode can continuously achieve PEC water splitting for over 520 h (Fig. 4g), the BVO-ΔOᵥ/FeOOH-PV artificial leaf is also expected to achieve hundreds hours of stability if the PV and connected circuit are encapsulated well.

Gas evolution performances of wireless artificial leaves with dimensions up to 3 cm × 3 cm were measured in a sealed reactor connecting to a GC (Supplementary Fig. 40). Larger area artificial leaves with dimensions of 6 cm × 6 cm (exposed area: 25 cm$^2$), 9 cm × 9 cm (exposed area: 56.25 cm$^2$), 12 cm × 12 cm (exposed area: 100 cm$^2$) and 21 cm × 21 cm (exposed area: 306.25 cm$^2$) were sealed in home-made quartz reactors with dimensions of 15 cm ×15 cm ×10 cm and 30 cm × 30 cm × 10 cm (Supplementary Fig. 41), and the produced gas samples were taken out with a syringe every 30 min, and injected in GC for testing. The gas production performance of 3 wireless artificial leaves were measured, and the average H$_2$ and O$_2$ evolution performances with error bars are shown in Supplementary Fig. 42. It can be observed that the performance of the wireless artificial leaf is similar to that of the wired artificial leaf (Fig. 5f). Based on the gas evolution performances (Supplementary Figs. 42, 43) and Supplementary Equation 19, the STH efficiencies of the artificial leaves with exposed areas of 0.126 cm$^2$, 1 cm$^2$, 6.25 cm$^2$, 25 cm$^2$, 56.25 cm$^2$, 100 cm$^2$, and 306.25 cm$^2$ can be calculated as 8.4%, 8.2%, 7.2%, 6.3%, 5.0%, 4.2%, and 2.7% (Supplementary Fig. 44). It can be observed that the STH efficiencies calculated from the production of hydrogen are slightly lower than those calculated from the operating photocurrent densities, which is because the Faradaic efficiency for PEC water splitting is not 100%. As shown in Supplementary Fig. 45a, the STH efficiencies of our developed BVO-ΔO$_v$/FeOOH-PV artificial leaves are the best amongst all BiVO$_4$-PV based unbiased water splitting systems[10,24,33–54]. The stability of our artificial leaf is also comparable to other state-of-the-art devices (Supplementary Fig. 45b and Supplementary Table 9).

It can be observed that the STH efficiency of the device decreases with the increase of size (Supplementary Fig. 44), which is caused by many factors such as ohmic losses in the FTO substrate, the electrolyte conductivity, and mass-transport (H$^+$/OH$^-$ ions) limitations[48]. The ohmic loss issue can be addressed by embedding metal gridlines (e.g., Ag, Ni) on FTO substrates[48,55]. In our work, the large-area photoanode was assembled by sub-photoanodes (dimensions: 3 cm × 3 cm) connected by conductive gridlines, which is also effective in reducing the ohmic losses within the film[56]. The electrolyte conductivity can be enhanced by increasing the buffer concentration in the electrolyte[48]. The formation of proper through-holes in the FTO substrates would provide more paths for the transport of the H$^+$/OH$^-$ ions during PEC water splitting, which may be helpful to mitigate the mass-transport issues. However, the loading mass of the photoanode materials is also reduced, which would inevitably decrease the light absorption efficiency. In addition, high concentration acidic or alkaline electrolytes are beneficial for the transport of the H$^+$/OH$^-$ ions. However, it is a big challenge to achieve long-term stability for the device in such a harsh environment.

For standalone artificial leaves, the production of a mixture of H$_2$ and O$_2$ gases may cause safety issues and requires the subsequent separation of H$_2$ and O$_2$. This issue can be addressed by assembling the wired artificial leaf in an H-type reactor with two chambers (denoted as Chamber 1 and Chamber 2) separated by a Nafion membrane. Specifically, the encapsulated BVO-ΔO$_v$/FeOOH-PV photoanode is placed in Chamber 1, while a Pt counter electrode is placed in Chamber 2. The BVO-ΔO$_v$/FeOOH-PV photoanode and the Pt counter electrode is connected through the external circuit. Under light illumination, O$_2$ can be collected in Chamber 1, whereas H$_2$ can be collected in Chamber 2. In this design, we can collect the separated H$_2$ and O$_2$ gases, respectively. It should be mentioned that the protonic conductivity of a Nafion membrane increases significantly with temperature (<80 °C)[57]. Since solar water splitting is generally operated at room temperature, the performance of this configuration may be limited by the resistance of the Nafion membrane[58]. On the other hand, membrane-free devices with liquid- or solid-state redox mediators can generate H$_2$ and O$_2$ gases in separated cells during water splitting[59,60], which can eliminate the resistance caused by the incorporation of membranes. However, owing to the increase distance between the two electrodes, the ohmic losses of the system also increase. More works are required to confirm the reliability of membrane-free devices for scale-up applications.

Life cycle assessment (LCA) has been widely applied to measure environmental sustainability among various products and systems[61,62]. In this study, we applied LCA using the openLCA (version 2.2.0) software to compare the environmental impacts of photoelectrochemical water splitting (Scenario 3) with natural gas reforming (Scenario 1) and electrocatalytic water splitting (Scenario 2) for H$_2$ production (Fig. 5g, h). The analysis was calculated on the basis of the production of 1 kg of H$_2$. The energy (e.g., electricity, natural gas, oil, fuel, and gas, etc.), materials (e.g., land, infrastructure, diesel, photovoltaic solar panels, solid materials (98% concrete + 2% supporting sand), and water), chemicals, and emissions during the production and transport of hydrogen in the three scenarios were taken into account. Detailed contributions of each category to the environmental impacts of three scenarios are listed in Supplementary Tables 10–16. Compared to natural gas reforming, both electrocatalytic water splitting and photoelectrochemical water splitting have little impact on climate change (6.15% and 1.13%, respectively). The global warming potential (GWP, kg$_{CO_2eq.}$kg$_{H_2}^{-1}$) of a H$_2$ production technology is also important to assess the environmental impacts[58]. The GWP values of natural gas reforming (Scenario 1), electrocatalytic water splitting (Scenario 2), and photoelectrochemical water splitting (Scenario 3) are 141.3, 8.7, and 2.1 kg$_{CO_2eq.}$kg$_{H_2}^{-1}$, respectively. In addition, electrocatalytic water splitting (Scenario 2) has significant impact on other environmental categories in terms of energy and chemicals during hydrogen production. Photoelectrochemical water splitting (Scenario 3) exhibits significant decreases across all environmental impact categories, indicating the environmentally friendly feature for H$_2$ production. Therefore, the development of photoelectrochemical water splitting technology is essential to realizing green H$_2$ production. In addition, the cost of producing 1 kg of H$_2$ using the above three technologies were also compared. As shown in Supplementary Fig. 46, the cost of photoelectrochemical water splitting is higher than natural gas reforming while lower than electrocatalytic water splitting. However, if taking the carbon tax into account, photoelectrochemical water splitting is competitive for hydrogen production in the future market.

## Discussion

In conclusion, we have developed BiVO$_4$ photoanodes with gradient distributed oxygen vacancies by converting from the electrodeposited Bi$_{34.7}$O$_{36}$(SO$_4$)$_{16}$ precursor films. Owing to the gradient distribution of oxygen vacancies within the BiVO$_4$ film, the electronic properties and other optoelectrical properties can be tailored, leading to a significantly enhanced charge separation efficiency. By loading with a FeOOH OEC to accelerate surface oxygen evolution kinetics, a photocurrent density of 7.0 mA cm$^{-2}$ at 1.23 V vs. RHE under AM 1.5 G illumination is achieved, which is 93.3% of its theoretical maximum. A charge separation efficiency of nearly 100% and a charge transfer efficiency of 94.9% are achieved. In addition, the optimized BiVO$_4$/FeOOH photoanode exhibits a long-term stability up to 520 h for PEC water splitting. Systematic studies reveal that the gradient distribution of oxygen vacancies in BiVO$_4$ films can effectively tune the electronic structure, and improve charge mobility, which significantly promote charge separation within the film, thus achieving a high photocurrent density. An artificial leaf composed of the BiVO$_4$/FeOOH photoanode and a silicon solar cell achieves an unbiased STH efficiency of 8.4%, with a long-term stability exceeding 50 h. Furthermore, a large-area artificial leaf with dimensions of 21 cm × 21 cm can generate observable hydrogen and oxygen bubbles under natural sunlight illumination, exhibiting an STH efficiency of 2.7%. LCA analysis shows that the PEC water-splitting process has a much smaller environmental footprint in terms of carbon emissions, chemical and energy consumption. Thus,

the PEC water-splitting technology shows great potential for scale-up sustainable hydrogen production.

# Methods

## Materials

Fluorine-doped tin oxide (FTO) substrates (F:SnO$_2$, 14 $\Omega$ per square, Yingkou OPV Tech New Energy Technology Co., Ltd), bismuth nitrate pentahydrate (Bi(NO$_3$)$_3$·5H$_2$O, Sigma-Aldrich, >98%), Thiourea (CH$_4$N$_2$S, Innochem, 99%), vanadyl acetylacetonate (VO(acac)$_2$, Sigma-Aldrich, 99%), nitric acid (HNO$_3$, Adamas, 69%), acetic acid glacial (CH$_3$COOH, Greagent, ≥99.8%), ferrous sulfate heptahydrate (FeSO$_4$·7H$_2$O, Sigma-Aldrich, 99%), nickel sulfate hexahydrate (Ni(SO$_4$)$_2$·6H$_2$O, Sigma-Aldrich, 99%), cobalt sulfate heptahydrate (CoSO$_4$·7H$_2$O, Sigma-Aldrich, 99%), acetone (Greagent, GR), ethanol (Greagent, ≥99.7%, GR), dimethyl sulfoxide (DMSO, Fisher, 99.9%), potassium iodide (KI, Alfa, ≥95%) and p-benzoquinone (Alfa, ≥98.0%), silicon solar cell panels with dimensions of 1 cm × 1 cm, 3 cm × 3 cm, 6 cm × 6 cm, 9 cm × 9 cm, 12 cm × 12 cm, and 21 cm × 21 cm (Risym, 1.5 V).

## Synthesis of BVO-$\Delta O_v$-x, BVO, and BVO-Ref photoanodes

The precursor solution was fabricated by dissolving 0.48 g of Bi(NO$_3$)$_3$·5H$_2$O in 50 mL of a 0.5 M thiourea solution. Nitric acid with a concentration of 69% was added drop-by-drop to achieve a pH of 1.7, followed by brief agitation. Electrodeposition was applied within a standard three-electrode cell configuration. An FTO substrate was used as the working electrode (WE), a saturated Ag/AgCl electrode was used as the reference electrode (RE), and a Pt electrode (purchased from Shanghai Chenhua Instrument Co., Ltd.) was used as the counter electrode (CE). To optimize the quality of the BSO precursor films, various deposition potentials (−0.3, −0.5, and −0.7 V vs Ag/AgCl) were applied. The obtained bismuth precursor films were denoted as BSO-3, BSO-5, and BSO-7, respectively. Based on the optimized potential, the electrodeposition time was also adjusted in the range of 1-4 min. After electrodeposition, the obtained bismuth precursor films were air-dried at room temperature for 30 min. A vanadium source solution was prepared by dissolving 1 mmol of vanadyl acetylacetonate in 5 mL of methanol with the assistance of ultrasonication. Then, 70 μL of the obtained vanadium source solution was uniformly distributed onto the BSO films, followed by annealing in air at 500 °C for 2 h, with a heating rate of 2 °C min$^{-1}$. This thermal process facilitated the full conversion of BSO into BiVO$_4$. After the thermal process, any remaining V$_2$O$_5$ on the BiVO$_4$ surface was eliminated via immersing in a 1 M NaOH solution for 30 min. The resultant pure BiVO$_4$ films were rinsed with deionized water and air-dried at room temperature. The BiVO$_4$ films converted from the BSO precursor films obtained by various deposition potentials (−0.3, −0.5, and −0.7 V vs Ag/AgCl) were denoted as BVO-$\Delta O_v$-3, BVO-$\Delta O_v$-5, and BVO-$\Delta O_v$-7, respectively. For comparison, another BiVO$_4$ sample denoted as BVO, was produced by replacing the nitric acid with glacial acetic acid while keeping the identical methodology and conditions.

To show the advancement of the BiVO$_4$ photoanodes converted from our developed bismuth precursor films, another BiVO$_4$ sample (BVO-Ref) was also prepared according to a previous report by Kim et al.[63] Briefly, HNO$_3$ was added drop-by-drop in 50 mL of a 0.4 M KI solution until the pH reached 1.7, followed dissolving 0.02 M of Bi(NO$_3$)$_3$ with the assistance of ultrasonication. Then, 20 mL of a 0.23 M p-benzoquinone ethanol (99.5%) solution was added to the above solution with stirring. BiOI films were prepared using a three-electrode system similar to the preparation of BSO precursor films. The applied potential was set as −0.1 V vs Ag/AgCl, and the deposited time was 4 min. The following BiVO$_4$ conversion process was the same as the BVO-$\Delta O_v$-5 films as described above to obtain BVO-Ref films with almost the same film thickness as BVO-$\Delta O_v$-5.

## Synthesis of BVO-$\Delta O_v$/OEC photoanodes

According to a previous report[63], the photo-assisted electrodeposition process of a FeOOH layer was carefully optimized to achieve the uniform growth and complete coverage of the FeOOH co-catalyst particles on the BiVO$_4$ particles. Specifically, a FeSO$_4$·7H$_2$O solution with a concentration of 0.01 mol L$^{-1}$ was prepared in deionized water previously purged with nitrogen for 30 min to prevent oxidization of Fe$^{2+}$. Photo-assisted electrodeposition was performed within a three-electrode cell consisting of a BiVO$_4$ film, a platinum wire, and an Ag/AgCl reference electrode. Illumination was facilitated by a 300 W xenon arc lamp equipped with an AM 1.5 G filter. Light was directed through the FTO substrate, and the light intensity at the FTO surface was carefully calibrated to 100 mW cm$^{-2}$. An external voltage of 0.25 V vs. Ag/AgCl was applied for 30 min, which was denoted as one deposition cycle. To achieve the optimized activity and stability, three consecutive deposition cycles were performed. To avoid the oxidation of Fe$^{2+}$ into Fe$^{3+}$, a newly prepared FeSO$_4$·7H$_2$O solution was used for each deposition cycle.

BVO-$\Delta O_v$/NiOOH, BVO-$\Delta O_v$/NiFeOOH, and BVO-$\Delta O_v$/NiFeCoOOH photoanodes were fabricated using a similar photo-assisted electrodeposition process as BVO-$\Delta O_v$/FeOOH. The electrolyte for the fabrication of BVO-$\Delta O_v$/NiOOH is 0.01 mol L$^{-1}$ of Ni(SO$_4$)$_2$·6H$_2$O. The electrolyte for the fabrication of BVO-$\Delta O_v$/NiFeOOH is composed of 0.005 mol L$^{-1}$ of Ni(SO$_4$)$_2$·6H$_2$O and 0.005 mol L$^{-1}$ of FeSO$_4$·7H$_2$O. The electrolyte for the fabrication of BVO-$\Delta O_v$/NiFeCoOOH is composed of 0.003 mol L$^{-1}$ of Ni(SO$_4$)$_2$·6H$_2$O, 0.005 mol L$^{-1}$ of FeSO$_4$·7H$_2$O, and 0.002 mol L$^{-1}$ of CoSO$_4$·7H$_2$O.

## Fabrication of large BiVO$_4$ photoanodes

BiVO$_4$ films with size of 3 cm × 3 cm were fabricated on FTO substrates using the same process of fabricating BVO-$\Delta$Ov. Subsequently, 4, 9, 16, and 49 pieces of the obtained BiVO$_4$ films with dimensions of 3 cm × 3 cm were assembled as the 2 × 2, 3 × 3, 4 × 4 and 7 × 7 square array, respectively. To ensure the excellent conductivity amongst all BiVO$_4$ films, the edge of each BiVO$_4$ film with a width of 2.5 mm was covered by silver conductive paint, followed by the coverage of a conductive graphite tape with a width of 5 mm to connect all BiVO$_4$ films (as shown in Supplementary Fig. 31). Then, large area BiVO$_4$ photoanodes with sizes of 36 cm$^2$, 81 cm$^2$, 144 cm$^2$, and 441 cm$^2$ (the area of the conductive graphite tape was included) were obtained.

## Fabrication of BVO-$\Delta O_v$/FeOOH−PV artificial leaves

The artificial leave was fabricated using BVO-$\Delta O_v$/FeOOH photoanodes and a photovoltaic (PV) panel. The PV panel consisted of a silicon solar cell, a water-proof epoxy resin sealing layer, and a PCB substrate, with metal contacts exposed on the back. PV panels with different sizes of 1 cm × 1 cm, 3 cm × 3 cm, 6 cm × 6 cm, 9 cm × 9 cm, and 21 cm × 21 cm were purchased from Kailin Electronic Technology Co., Ltd. The edges of photoanodes with different sizes were pasted on the front of the PV panels with the same sizes. The multiple positive electrode contacts on the back of the PV panel were connected to the conductive graphite tapes on the adjacent photoanode surface through conductive silver paste. The negative electrode on the back of the PV panel was welded with a platinum wire. All exposed metal contacts and conductive silver paste on the back of the PV panel were encapsulated with a water-proof silicone coating to prevent water and air from corroding the circuit. The photoanode was connected with the PV panel in tandem. Supplementary Figs. 33, 34 show the connection scheme of a 6 cm × 6 cm (exposed area: 25 cm$^2$) device, and the digital images of 1 cm × 1 cm (exposed area: 1 cm$^2$), 3 cm × 3 cm (exposed area: 6.25 cm$^2$), 6 cm × 6 cm (exposed area: 25 cm$^2$), 9 cm × 9 cm (exposed area: 56.25 cm$^2$), and 21 cm × 21 cm (exposed area: 306.25 cm$^2$) devices.

## Characterizations

Crystal structures of the samples were characterized by X-ray diffraction (Bruker D8 Advanced PXRD, l ¼ 1.5418 Å, 298 K, Ni-filtered Cu Ka-radiation). The morphology and microstructure of the samples were investigated by transmission electron microscopy (TEM, FEI Talos F200X) and field-emission scanning electron microscopy (FE-SEM, FEI Verios G4). The lattice structures of the samples were observed by high-resolution transmission electron microscopy (HRTEM, FEI Talos F200X). The light absorption properties of the samples were measured using an ultraviolet-visible (UV-Vis) spectrophotometer (Shimadzu UV-2600i). X-ray absorption fine spectroscopy (XAFS) data for the Bi $L$3-edge and V $K$-edge of the samples were collected on Beamline BL14W1 at the Shanghai Synchrotron Radiation Facility (SSRF) using a transmission mode and a fluorescent mode, respectively. X-ray photoelectron spectroscopy (XPS) spectra of the samples were recorded using a K-Alpha X-ray photoelectron spectrometer (Thermo Scientific Inc.) equipped with a monochromatic Al Kα line as the X-ray source. The binding energies were calibrated with respect to the residual C 1 s peak at 284.8 eV. The photoluminescence (PL) spectra (excited by a 370 nm light illumination) were measured using a fluorescence spectrophotometer (FLS100, Edinburgh). The time-resolved photoluminescence (TRPL) curves of the samples were measured on a fluorescence lifetime spectrophotometer (FLS100, Edinburgh) under the excitation of a 370 nm laser.

## PEC performance measurements

PEC measurements were carried out on an electrochemical workstation (CHI 760E) at room temperature using a conventional three-electrode cell. The prepared photoanodes were employed as the WE, while Ag/AgCl electrode and Pt electrode were used as the RE and CE, respectively. The light source was a Xe 300 W lamp (CEL-S300, CEAULIGHT) with an AM 1.5 G filter, and the light intensity at the WE was calibrated to 100 mW cm$^{-2}$ using an optical power meter (CEL-NP2000-2A, CEAULIGHT). The spectrum of the light source was confirmed to match the standard AM 1.5 G spectrum using a MAX2000-Pro spectroradiometer (Shanghai Wyoptics Technology Co., Ltd). The exposed area of the photoanode was 0.126 cm$^2$. The back-side illumination through the FTO side was adopted for all the PEC tests. 1 M potassium borate buffers with and without 0.2 M of $Na_2SO_3$ (pH=9.5) were used as the electrolyte.

## $H_2$ and $O_2$ evolution measurements

As shown in Supplementary Figs. 38 and 40, $H_2$ and $O_2$ evolution performance of the wired and wireless BVO-$\Delta O_V$/FeOOH–PV artificial leaves with dimensions up to 3 cm × 3 cm were measured in a photocatalytic activity evaluation system (Beijing China Education Au-Light Co., Ltd., CEL-PAEM-D8) connected with a gas chromatography (Beijing China Education Au-Light Co., Ltd., GC-7920). A Xe 300 W lamp (CEL-S300, CEAULIGHT) equipped with an AM 1.5 G filter was used as the light source. The light intensity at the artificial leaf was carefully calibrated to 100 mW cm$^{-2}$ with the spectrum matching the standard AM 1.5 G spectrum. A 1 M potassium borate buffer (pH=9.5) was used as the electrolyte for all measurements. The produced gases were analyzed every 30 min or 1 h automatically controlled by the software.

Wireless artificial leaves with dimensions of 6 cm × 6 cm (exposed area: 25 cm$^2$), 9 cm × 9 cm (exposed area: 56.25 cm$^2$), 12 cm × 12 cm (exposed area: 100 cm$^2$) and 21 cm × 21 cm (exposed area: 306.25 cm$^2$) were sealed in home-made quartz reactors with dimensions of 15 cm ×15 cm ×10 cm and 30 cm × 30 cm ×10 cm (Supplementary Fig. 41), and the produced gas samples were taken out with a syringe every 30 min, and injected in GC every for testing.

## Computational method

First-principles calculations based on density functional theory were using VASP[64], where the Perdew-Burke-Ernzerh of generalized gradient approximation (GGA) method was applied[65]. The cut-off energy for plane wave basis was set to 450 eV. A Γ-centered Monkhorst-Pack 4 × 4 × 1 k-mesh was adopted to sample the Brillouin zone[66]. The convergence criteria were set to be 1.0 ×10$^{-5}$ eV and 0.01 eV Å$^{-1}$ per atom. All the systems were fully relaxed before the calculations of electronic structures, in which spin–orbit coupling effects were included.

## Life-cycle assessment analysis

This study applied life cycle assessment (LCA) method to compare the environmental impacts of three different hydrogen production methods. The production of 1 kilogram hydrogen was the function unit and all the energy consumption, materials use, and emissions were referred to the function unit.

(a) Hydrogen production by natural gas reforming, the data for natural gas extraction, processing and transportation were from ref. 67, the data for steam reforming were from ref. 68; (b) Electrocatalytic hydrogen production, using a commonly used alkaline electrolyzer, and the data source was from ref. 69; (c) Hydrogen production by photoelectrochemical water splitting, and the data source was from the proportional amplification of the experimental process of this work. The system boundary included the extraction and processing of raw materials, energy and materials use, transportation and emissions. The foreground inventory data were mainly obtained from experiments and literature, including energy and chemical consumption and exhaust emissions, while the background inventory data were obtained from the Ecoinvent3.8 database. The entire modeling process was carried by OpenLCA (version 2.0.4).

The life-cycle environmental effect results were calculated using the ReCiPe method. This method was one of the commonly used methods for a wide application of LCA analysis. The environmental impact categories considered for the LCA of each process and energy included climate change, agricultural land occupation, fossil depletion, freshwater ecotoxicity, freshwater eutrophication, human toxicity, ionising radiation, marine ecotoxicity, marine eutrophication, metal depletion, natural land transformation, ozone depletion, particulate matter formation, photochemical oxidant formation, terrestrial acidification, terrestrial ecotoxicity, urban land occupation, and water depletion.

## Data availability

The data generated within the main text and its Supplementary Information file are available in the Source Data file. Source data are provided with this paper.

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

## Acknowledgements

The authors would like to acknowledge the financial support from the National Natural Science Foundation of China (No. 52372292 (S.W.), 62288102 (W.H.)), Shenzhen Science and Technology Program (No. JCYJ20220530161615035 (S.W.), JCYJ20240813150835046 (S.W.)), the Fundamental Research Funds for the Central Universities (S.W.), and material characterizations from the Analytical & Testing Center of Northwestern Polytechnical University.

## Author contributions

S.W. guided and designed the project. B.L. carried out the experiments and wrote the manuscript. X.W. and Y.Z. conducted SEM characteriza-tion and assisted in the large area artificial leaf fabrication and perfor-mance measurement. M.Z. gave suggestions and comments on the artificial leaf fabrication. C.Z. and S.L. conducted life cycle assessment (LCA) analysis. Y.M. conducted HRTEM characterization and analysis. B.L., S.W., M.Z., and W.H. analyzed the data and prepared the manu-script. All the authors commented and approved the paper.

## Competing interests

The authors declare no competing interests.

## Additional information

**Peer review information** *Nature Communications* thanks Jae Sung Lee, Kazunari Domen and the other, anonymous, reviewer for their con-tribution to the peer review of this work. A peer review file is available.

