## [Peer Review File · Nature Communications]

A standalone bismuth vanadate-silicon artificial leaf achieving 8.4% efficiency for hydrogen production

Corresponding Author: Professor Songcan Wang

Version 0:

Reviewer comments:

Reviewer #1

(Remarks to the Author)

Authors report the ultrastable BiVO₄ photoanode and application for large-area artificial leaf with solar-to-hydrogen efficiency of 8.8%. Authors put detailed study of BiVO₄ component with useful information for high PEC and stable performance, keeping the information very few about scale-up. However, this work could be useful for scalable PEC water splitting. I have following suggestions with necessary comments/information before recommending to publish in the esteem journal.

- 1) In introduction, author discuss only III-V class photoelectrodes but they should include halide perovskite PEC devices which are efficient and scable also.
- 2) Author report "new electrodeposition process" for BiVO₄, the electrodeposition process is already known method even for BiVO₄, author should highlight the main new point which is not reported yet.
- 3) Author must add the fabrication steps and methodology of large BiVO₄ photoanodes including how 7 x 7 array designed fabricated, how gridlines were fabricated? These procedure is missing in the method section.
- 4) Author use FeOOH as OER catalyst, however, there are research on effect of NiOOH, NiFeOOH and NiFeCo which are also good candidates. Did author check such catalyst to optimize the performance ??
- 5) Why author measured stability at 0.6VRHE, is it due to high ABPE at this voltage? When and why the device degraded after 500 hours? Did author also measured stability at 1.23VRHE?
Is 500 hours a maximum period of time mesasured for photoanode?. If device was degraded even after then put the data and explain the reason of degradation beyond 500h.
- 6) What is actual area of BiVO₄ and Si PV component in PEC-PV system for Figure 5e? Are both area kept same? Which configuration was used , tandem or parallel in all cases??
Similarly, what is the are pf PEC-PV system in graph Figure 5g?
Explain all in caption and methodology.
- 7) Technical specification of Si PV devices are missing, include all infomration such as J_{sc}, Voc, FF, PCE of Si PV devices with small solar cells and large-modules. In Photograph, readers can not see Si PV Panel. Keep photographs of BiVO₄ and Si PV panel properly, how the both BiVO₄ and Si PV panel were arranged and connected in PEC-PV system?
- 8) What is the gridlines indicate in Figure 5a, 5d?
Also Figure 5 rearrangement is necessary as per follow,
Old Figure 5b (both schematics) Figure 5a
Old Figure 5c Figure 5b
Old Figure 5a Figure 5c
Old Figure 5g Figure 5f
Old Figure 5h Figure 5g
Old Figure 5f Figure 5h
- 9) Considering the title of the manuscript, author should add the studying the geometrical area effect of PEC devices. i.e., J-V performance of different size (0.126 cm², 1 cm², 9 cm², 36 cm², 100 cm² and 440 cm²) of BiVO₄ photoanodes.
- 10) STH efficiency of 8.8 % of which size of PEC-PV system?
Discuss STH efficiency of above varying the size PEC-PV systems and gas evolution rate for each.
Effect of area on STH efficiency performance of PEC-PV system, generally STH efficiency decrease while increasing the area.
3 x 3 (9 cm²) PEC-PV system

6 x 6 (36 cm²) PEC-PV system
10 x 10 (9 cm²) PEC-PV system
21 x 21 (440 cm²) PEC-PV system

Author should can add BiVO₄-Si PV cell as wired or wireless artificial device. Wired device can help to measured J-V curve + stability + gas productivity, while wireless artificial device can only provide gas productivity. So author should compare in the way to provide useful information systematically. Figure 5 neither clarifiy nor represent anything.

11) How the gas productivity was measured of artificial leaf, did author used specific PEC reactor with membrane , please provide the data of PEC cell reactor.

12) Is this 50 hours a maximum period of time mesasured for PEC-PV artificial leaf?. If device was degraded even after then put the data and explain the reason of degradation beyond 50h.

13) For LCA study, author should state couple sentence about global warming potential (GWP), as new green hydrogen production technology should not affect the global warming much compared to existing technologies such steam methane reforming (SMR) and other green hydrogen production technology. Refer paper : Chem. Soc. Rev., 53, 2388-2434, 2024.

14) In the attempt of scalability and LCA of PEC system, author should brieflt discuss the techno-economic part. e.g., Cost of Photoanode device and PEC system.

15) Author should cite the sclable PEC papers (Nat. Energy 9, 272–284, 2024; Energy Environ. Sci.,17, 3604-3617, 2024; J. Power Sources, 398, 224, 2018; J. Power Sources, 454, 2020; Sustainable Energy Fuels, 8, 3726-3739; 2024; Small Methods, 7, 10, 2300619, 2023) in main script and compare the STH efficiency in table in Supplementary information.

Reviewer #2

(Remarks to the Author)

In the present manuscript the authors study a BiVO₄ photoanode to fabricate an artificial leaf coupled with a Si-based PV cell. Along this study, the authors performed a detailed structural, morphological, chemical and photoelectrochemical characterization of the employed materials through a wide range of different techniques. Additionally, a scaled-up artificial leaf has been successfully developed and tested under relevant working conditions. The manuscript is well written and the experiments well performed.

However, even the manuscript is of the true interest of the solar fuels community and it is well aligned with the scope of Nature Communications, there are some major points that have to be addressed before publication in order to achieve the high-quality standards required in this prestigious journal.

1. The quality and contrast of the SEM images shown in Figure 1 need to be improved.
2. The authors need to explain better what is the difference between BVO-5 and BVO-5a. It is not clear for the reader how it is written leading to confusion between thermal treatments and electrolyte modifications (between nitric and acetic acid).
3. The authors have to report the photocurrent densities, IPCE and the rest of the PEC characterisation without hole scavenger to ease the comparison and benchmark of this work with other previous works on the state-of-the-art. Basically, Figure 2 has to be repeated in bare aqueous electrolyte without hole scavenger. On this regard, why the authors employed borate buffer instead of KPI buffer? Usually, charge separation efficiency versus applied potential (Figure 2c) is represented in continuous line.
4. Why the authors did not check the Vanadium edge on XAS instead of the Bi one? The V is the more sensitive edge to oxygen vacancies. Please take a look to the following reference and add to the manuscript:
<https://doi.org/10.1021/acscami.2c07451>.
5. The authors should elaborate more in detail the structure and components of the artificial leaf. I strongly recommend the authors to include a detailed scheme of the different components.

Reviewer #3

(Remarks to the Author)

In this study, the authors present a compelling design for a BiVO₄ (BVO) photoanode characterized by a gradient distribution of oxygen vacancies, which generates strong dipole fields within the film to enhance charge separation. Notably, the device achieved a remarkable photocurrent density of approximately 7 mA/cm² at 1.23 V versus the reversible hydrogen electrode (RHE), as well as exhibiting stability over a 500-hour duration. Additionally, by integrating the fabricated photoanode with a silicon solar cell in a tandem configuration, the authors achieved a solar-to-hydrogen (STH) conversion efficiency of 8.8%. Furthermore, they demonstrated the feasibility of scaling up the BVO photoanode and its integration with a silicon solar cell with an area of approximately 441 cm². The structural and performance evaluations are substantively supported by experimental data. However, several significant comments and suggestions must be addressed before the manuscript can be recommended for publication:

1. The title of the manuscript requires revision. The authors demonstrated an STH conversion efficiency of approximately 8.8% using a small BVO photoanode with an area of 0.126 cm², rather than from a large-area BVO photoanode. It is therefore advisable for the authors to reconsider the title for clarity and accuracy.
2. Figure 2b: The reviewer recommends that the authors provide the estimated photocurrent for the photoanodes depicted in the graph, determined from the integrated IPCE spectra over the standard solar spectrum, to corroborate the photocurrent

data presented in Figure 2a.

3. Page 18, lines 317-318: The authors should amend the phrase "To further evaluate the overall water splitting..." to "To further evaluate water splitting ...".

4. Figure 4h: The authors are encouraged to calculate the theoretical generation rates of O₂ and H₂ and to determine the Faradaic efficiency of the gas evolution reactions. Furthermore, the units for the evolved gases in the graph should be expressed as $\mu\text{mol}/\text{cm}^2$.

5. The authors must present the data on photoelectrochemical (PEC) evaluation more clearly. It remains unclear which irradiation source was utilized in this study. Was all data obtained using a Xe lamp with a 1.5 AM filter, or was a solar simulator employed? The reviewer suggests that the authors provide additional details regarding the irradiation source and elaborate on the calibration procedure for this source. Ideally, the authors should present the calibrated spectrum of the source used in this investigation and compare it with the AM 1.5G solar irradiance spectrum at various wavelengths.

6. In Figure 4g, the authors report results from stability tests conducted at a potential of 0.6 V_{RHE} under simulated solar conditions. It appears that the temperature of the electrolyte was not maintained at a constant level during these stability tests. The observed increase in photocurrent from the beginning of the experiment may be attributed to thermal heating of the electrolyte due to solar infrared radiation. The authors are advised to address this potential concern.

7. Figure 5e: The authors report STH value derived from intersection points and photocurrent density using a specific equation. The STH efficiency should be measured with respect to the V_{RHE} scale but should also account for measurements in a two-electrode system relevant to the V_{CE} scale. To accurately determine the real STH value, the authors should calculate the STH efficiency based on gas evolution using Faraday's law of electrolysis. This value will represent the measured STH efficiency of the tandem device.

8. Page 20: In this section, the authors discuss the fabrication of a large-scale tandem device with a surface area of approximately 441 cm² (Figure 5d) and demonstrate its operational principle under natural sunlight. However, it would be preferred if the authors could provide the actual STH efficiency of the larger tandem device when operating under natural sunlight. Furthermore, should the present system be scaled up to a panel size of 1 m², the fabricated device is expected to produce a mixture of hydrogen and oxygen gases. This raises concerns about the effective separation of hydrogen from the proposed system. It would be advantageous for the authors to address this issue, detailing the methodologies or technologies that could be implemented to facilitate the efficient separation of hydrogen.

9. Figure 5h: The data presented concerning the H₂/O₂ evolution from the BVO-PV tandem device raises several questions for the reviewer, particularly regarding the rationale for replacing the electrolyte every 10 hours during the testing procedure. If the device requires fresh electrolyte at this interval, it could present significant challenges for scaling up the system for practical applications. Additionally, it appears that the rates of H₂ and O₂ production decrease throughout the testing period. The authors are encouraged to address these concerns.

10. In Figures 5g and h, the unit for photocurrent density should be specified in mA/cm², and the unit for evolved gases must be indicated as $\mu\text{mol}/\text{cm}^2$.

Version 1:

Reviewer comments:

Reviewer #1

(Remarks to the Author)

Authors arranged the revision part exceptionally well based on my previous comments and now manuscript seems suitable for publication in the current form. With following minor suggestion in the main manuscript, it is advisable to add one sentence and citing relevant proper references for origin of gridline design ideas in photoelectrodes.

Joule, 7, 884–919, 2023;
Chem. Soc. Rev., 53, 2388-2434, 2024;
Int. J. Hydrogen Energy 36, 52625270, 2011;
J. Power Sources, 398, 224-232, 2018;
J. Power Sources, 454, 227890, 2020;
Sustainable Energy Fuels, 8, 3726-3739, 2024;
Sustainable Energy Fuels, 3, 2366-2379, 2019

Reviewer #2

(Remarks to the Author)

Since the authors have extensively addressed all the comments and issues raised by the referees, I recommend publication of this manuscript in Nature Communications.

Reviewer #3

(Remarks to the Author)

I have completed the second revision of the manuscript titled "A Standalone Bismuth Vanadate-Silicon Artificial Leaf Achieving 8.4% Efficiency for Zero-Emission Hydrogen Production." Upon review, I note that the authors have addressed the referee's comments and suggestions with due diligence. I recommend the manuscript for publication in Nature Communications, pending the incorporation of the minor revisions detailed below.

1. Figure 4h: The current presentation of this figure appears overly complex. I suggest the authors consider removing the data corresponding to photocurrent measurements at 1.23 V versus RHE from this figure. Instead, this data could be effectively integrated into Figure 4g. Moreover, if the authors have conducted stability measurements of the photoanode at 1.23 V versus RHE over an extended duration, it would be beneficial to include this data in Figure 4g as well.

2. Methods Section: The authors are advised to specify the model of the light power meter employed to calibrate the Xe-lamp for clarity and reproducibility.

3. Regarding Previous Inquiry 8 (Oxygen and Hydrogen Separation): The authors propose the use of an H-type reactor with two chambers separated by a Nafion membrane for gas separation. However, it is important to note that the conductivity of the Nafion membrane is temperature-dependent, with proton exchange membrane devices typically operating near 80 °C. I encourage the authors to provide a more detailed discussion regarding the future development of large-scale systems that can efficiently separate hydrogen, including considerations for optimizing the operating conditions.

4. Supplementary Figure 14: This figure illustrates the solar-to-hydrogen (STH) efficiencies of wireless artificial leaves of varying device sizes. It is evident from the data that the STH efficiency is size-dependent, with a noted decrease to 2.7% when the device size reaches 306 cm². The authors are encouraged to provide a comprehensive explanation of the primary factors affecting STH efficiency as device size increases and to discuss potential strategies to mitigate this observed decline.

Version 2:

Reviewer comments:

Reviewer #3

(Remarks to the Author)

I have read the paper "A standalone bismuth vanadate-silicon artificial leaf achieving 8.4% efficiency for zero-emission hydrogen production" after final revision and I can clearly say that the authors revised the manuscript following all the referees' comments and suggestions. The current version of the manuscript can be recommended for publication in Nature Communications.

Response to Reviewers' Comments

Reviewer #1 (Remarks to the Author):

Authors report the ultrastable BiVO₄ photoanode and application for large-area artificial leaf with solar-to-hydrogen efficiency of 8.8%. Authors put detailed study of BiVO₄ component with useful information for high PEC and stable performance, keeping the information very few about scale-up. However, this work could be useful for scalable PEC water splitting. I have following suggestions with necessary comments/information before recommending to publish in the esteem journal.

Reply: Many thanks for your positive feedbacks of our work, and your constructive suggestions to further improve the quality of our work. We have carefully considered all of your suggestions and comments. Please read our point-to-point response and revision of our work as below.

1) In introduction, author discuss only III-V class photoelectrodes but they should include halide perovskite PEC devices which are efficient and scable also.

Reply: Thank you for your valuable comments. We totally agree that the performance of perovskite PEC devices have been significantly improved in recent years, which has driven PEC water splitting to a higher level. We are sorry for ignoring this kind of important device. In our revision, we have added the progress of perovskite PEC devices, as highlighted in Page 3.

Changes to the revised manuscript are shown below.

Main Manuscript (Introduction):

Page 3: Recently, perovskite PEC cells have shown great potential for efficient and scalable solar water splitting. For example, a monolithic stacked silicon-perovskite tandem exhibited a solar-to-hydrogen (STH) efficiency of 20.8%⁹. However, the photocurrent of the device degraded to 60% after 102 h of continuous operation. A Ni-encapsulated FAPbI₃ photoanode could stably split water for 72 h, and the assembled unbiased device with a size of 123 cm² achieved an STH efficiency of 8.5%¹⁰.

References:

9 Fehr, A. M. K. *et al.* Integrated halide perovskite photoelectrochemical cells with

solar-driven water-splitting efficiency of 20.8%. *Nat. Commun.* **14**, 3797 (2023).

10 Hansora, D. *et al.* All-perovskite-based unassisted photoelectrochemical water splitting system for efficient, stable and scalable solar hydrogen production. *Nat. Energy* **9**, 272-284 (2024).

2) Author report “new electrodeposition process” for BiVO₄, the electrodeposition process is already known method even for BiVO₄, author should highlight the main new point which is not reported yet.

Reply: Thank you for your suggestions. We are sorry for not expressing our idea accurately in our previous version. Yes, electrodeposition for BiVO₄ have been reported in many publications. Many parameters such as the recipe of the electrolyte, the applied potential, electrodeposition time and temperature, etc. will affect the quality of the BiVO₄ film. In this work, we would like to highlight that we have developed a new electrolyte recipe that has not been reported before, which can form Bi_{34.7}O₃₆(SO₄)₁₆ precursor films. After converting to BiVO₄ films, gradient distribution of oxygen vacancies across the film can be observed, which creates a strong built-in electric field that significantly promote charge transfer and separation in the bulk.

To avoid any misunderstanding and make the expression more accurate, we have changed “new electrodeposition process” as “electrolyte recipe”, as highlighted in Page 4 in the revised manuscript.

Changes to the revised manuscript are shown below.

Main Manuscript (Introduction):

Page 4: Here, we develop an electrolyte recipe for the preparation of Bi_{34.7}O₃₆(SO₄)₁₆ precursor films, followed by solid thermal reaction with vanadyl acetylacetonate (VO(acac)₂) to construct wormlike BiVO₄ photoanodes with a gradient distribution of oxygen vacancies (denoted as BVO-ΔO_v).

3) Author must add the fabrication steps and methodology of large BiVO₄ photoanodes including how 7 x 7 array designed fabricated, how gridlines were fabricated? These procedure is missing in the method section.

Reply: We appreciate your important suggestions. We apologize for missing the information for fabricating large BiVO₄ photoanodes and PEC-PV devices in the manuscript. We have added the required information in our revised manuscript, as highlighted in Page 33-34.

Changes to the revised manuscript are shown below.

Main Manuscript (Methods – Materials preparation)

Pages 33-34:

Fabrication of large BiVO₄ photoanodes

BiVO₄ films with size of 3 cm × 3 cm were fabricated on FTO substrates using the same process of fabricating BVO-ΔOv. Subsequently, 4, 9, 16 and 49 pieces of the obtained BiVO₄ films with dimensions of 3 cm × 3 cm were assembled as the 2 × 2, 3 × 3, 4 × 4 and 7 × 7 square array, respectively. To ensure the excellent conductivity amongst all BiVO₄ films, the edge of each BiVO₄ film with a width of 2.5 mm was covered by silver conductive paint, followed by the coverage of a conductive graphite tape with a width of 5 mm to connect all BiVO₄ films (as shown in Supplementary Fig. 32). Then, large area BiVO₄ photoanodes with sizes of 36 cm², 81 cm², 144 cm², and 441 cm² (the area of the conductive graphite tape was included) were obtained.

Fabrication of BVO-ΔOv/FeOOH–PV artificial leaves

The artificial leaf was fabricated using BVO-ΔOv/FeOOH photoanode and a photovoltaic (PV) panel. The PV panel consisted of a silicon solar cell, a water-proof epoxy resin sealing layer, and a PCB substrate, with metal contacts exposed on the back. PV panels with different sizes of 1 cm × 1 cm, 3 cm × 3 cm, 6 cm × 6 cm, 9 cm × 9 cm, and 21 cm × 21 cm were purchased from Kailin Electronic Technology Co., Ltd. The edges of photoanodes with different sizes were pasted on the front of the PV panels with the same sizes. The multiple positive electrode contacts on the back of the PV panel were connected to the conductive graphite tapes on the adjacent photoanode surface through conductive silver paste. The negative electrode on the back of the PV panel was welded with a platinum wire. All exposed metal contacts and conductive silver paste on the back of the PV panel were encapsulated with a water-proof epoxy resin coating to prevent water and air from corroding the circuit. The photoanode was connected with the PV panel in tandem. Supplementary Figs. 33-35 show the connection scheme of a 6 cm × 6 cm (exposed area: 25 cm²) device, and the digital

images of $1\text{ cm} \times 1\text{ cm}$ (exposed area: 1 cm^2), $3\text{ cm} \times 3\text{ cm}$ (exposed area: 6.25 cm^2), $6\text{ cm} \times 6\text{ cm}$ (exposed area: 25 cm^2), $9\text{ cm} \times 9\text{ cm}$ (exposed area: 56.25 cm^2), and $21\text{ cm} \times 21\text{ cm}$ (exposed area: 306.25 cm^2) devices.

Supplementary Fig. 32. Scheme of the assembly of large-area photoanodes. a An example of assembling a 2×2 array BVO- ΔO_v /FeOOH photoanode. **b** schemes of 3×3 , 4×4 , and 7×7 array BVO- ΔO_v /FeOOH photoanodes.

Supplementary Fig. 33. Scheme of the assembly of a 7 × 7 array artificial leaf. Illustration of a 21 cm × 21 cm Si PV panel, a BVO-ΔOv/FeOOH photoanode (21 cm × 21 cm), and different views of the artificial leaf.

Supplementary Fig. 34. The structure of a 2 × 2 array artificial leaf. Digital photos of a 2 × 2 array artificial leaf with different components.

Supplementary Fig. 35. Artificial leaves with different sizes. Digital photos of (a) a 21 cm × 21 cm artificial leaf, and (b) artificial leaves with different dimensions of 1 cm × 1 cm, 3 cm × 3 cm, 6 cm × 6 cm, 9 cm × 9 cm.

4) Author use FeOOH as OER catalyst, however, there are research on effect of NiOOH, NiFeOOH and NiFeCo which are also good candidates. Did author check such catalyst to optimize the performance?

Reply: Many thanks for your suggestions. We have prepared other samples with NiOOH, NiFeOOH and NiFeCoOH as the OER catalyst using a photo-assisted electrodeposition process, and the obtained samples were denoted as BVO- ΔO_v /NiOOH, BVO- ΔO_v /NiFeOOH, and BVO- ΔO_v /NiFeCoOOH, respectively. Their PEC water splitting performances were shown in Supplementary Figure 21a. It can be observed that the photocurrent densities of BVO- ΔO_v /NiOOH, BVO- ΔO_v /NiFeOOH, and BVO- ΔO_v /NiFeCoOOH at 1.23 V vs. RHE are 6.6, 6.8, and 7.1 mA cm⁻², respectively. Therefore, the BVO- ΔO_v /FeOOH sample prepared by our modified method with a special sea urchin structure shows comparable photocurrent densities with the champion BVO- ΔO_v /NiFeCoOOH. Since the NiFeCoOH OER catalyst contains different metal elements of Ni, Fe and Co, it is challenging to obtain the NiFeCoOH OER catalysts with exactly the same Ni: Fe: Co ratios in different batches. In addition, the photocorrosion of any element in the NiFeCoOH OER catalyst will cause the instability of the surface structure. Therefore, the long-term stability of BVO- ΔO_v /NiFeCoOOH is not good. Overall, the BVO- ΔO_v /FeOOH obtained in our work can achieve both high PEC activity and long-term stability.

Changes to the revised manuscript are shown below.

Main Manuscript (Results, Methods):

Page 18: Loading the photoanode surfaces with proper OEC is essential for efficient and stable PEC water splitting. The BVO- ΔO_v samples were decorated by FeOOH, NiOOH, NiFeOOH and NiFeCoOOH OECs using a photo-assisted electrodeposition process, and the obtained samples were denoted as BVO- ΔO_v /FeOOH, BVO- ΔO_v /NiOOH, BVO- ΔO_v /NiFeOOH, and BVO- ΔO_v /NiFeCoOOH, respectively. Although, BVO- ΔO_v /NiFeCoOOH exhibits a slightly higher photocurrent density than BVO- ΔO_v /FeOOH at 1.23 V vs. RHE, the stability is poor (Supplementary Discussion and Supplementary Fig. 21). Therefore, FeOOH is selected as the OEC for further investigation.

Page 32-33: BVO- ΔO_v /NiOOH, BVO- ΔO_v /NiFeOOH, and BVO- ΔO_v /NiFeCoOOH photoanodes were fabricated using a similar photo-assisted electrodeposition process as BVO- ΔO_v /FeOOH. The electrolyte for the fabrication of BVO- ΔO_v /NiOOH is 0.01 mol/L of $\text{Ni}(\text{SO}_4)_2 \cdot 6\text{H}_2\text{O}$. The electrolyte for the fabrication of BVO- ΔO_v /NiFeOOH is composed of 0.005 mol/L of $\text{Ni}(\text{SO}_4)_2 \cdot 6\text{H}_2\text{O}$ and 0.005 mol/L of $\text{FeSO}_4 \cdot 7\text{H}_2\text{O}$. The electrolyte for the fabrication of BVO- ΔO_v /NiFeCoOOH is composed of 0.003 mol/L of $\text{Ni}(\text{SO}_4)_2 \cdot 6\text{H}_2\text{O}$, 0.005 mol/L of $\text{FeSO}_4 \cdot 7\text{H}_2\text{O}$, and 0.002 mol/L of $\text{CoSO}_4 \cdot 7\text{H}_2\text{O}$.

Supplementary Information

Page 27: The BVO- ΔO_v samples were decorated by FeOOH, NiOOH, NiFeOOH and NiFeCoOOH OECs using a photo-assisted electrodeposition process, and the obtained samples were denoted as BVO- ΔO_v /FeOOH, BVO- ΔO_v /NiOOH, BVO- ΔO_v /NiFeOOH, and BVO- ΔO_v /NiFeCoOOH, respectively. As shown in Supplementary Fig. 21a, the photocurrent densities of the BVO- ΔO_v /FeOOH, BVO- ΔO_v /NiOOH, BVO- ΔO_v /NiFeOOH, and BVO- ΔO_v /NiFeCoOOH samples at 1.23 V vs. RHE are 7.0, 6.6, 6.8, and 7.1 mA cm^{-2} , respectively. Although the BVO- ΔO_v /NiFeCoOOH sample exhibits a lightly higher photocurrent density compared to BVO- ΔO_v /FeOOH, the photocurrent density of BVO- ΔO_v /NiFeCoOOH decreases from 6.6 to 4.6 mA cm^{-2} with a retention rate of 69.6% after 13 h of consecutive AM 1.5G illumination at 1.23 V vs. RHE (Supplementary Fig. 21b). Since the NiFeCoOH OEC contains different metal elements of Ni, Fe and Co, it is challenging to obtain the NiFeCoOH OEC with exactly the same Ni: Fe: Co ratios in different batches. In addition, the photocorrosion of any element in the NiFeCoOH OEC will cause the instability of the surface structure.

Therefore, the long-term stability of BVO- Δ O_v/NiFeCoOOH is not good.

Supplementary Fig. 21. PEC performance of the samples with different OECs. a LSV curves of the BVO- Δ O_v/FeOOH, BVO- Δ O_v/NiOOH, BVO- Δ O_v/NiFeOOH, and BVO- Δ O_v/NiFeCoOOH samples. **b** J-t curve of BVO- Δ O_v/NiFeCoOOH at 1.23 V vs. RHE. All measurements are in a 1 M borate buffer electrolyte (pH 9.5) under AM 1.5G illumination.

5) Why author measured stability at 0.6 V_{RHE}, is it due to high ABPE at this voltage? When and why the device degraded after 500 hours? Did author also measure stability at 1.23 V_{RHE}?

Is 500 hours a maximum period of time measured for photoanode? If device was degraded even after then put the data and explain the reason of degradation beyond 500h.

Reply: Thanks a lot for your insightful comments. Please allow us to provide detailed explanations of measuring the stability at 0.6 V_{RHE}. First of all, according to Fig. 4f, the FeOOH/BVO photoanode exhibits the highest ABPE value of 2.78% at around 0.6 V_{RHE}. Thus, the FeOOH/BVO photoanode can achieve the highest energy conversion efficiency (considering the energy input of both light and electricity) at 0.6 V_{RHE}. Secondly, when combining a photoanode and a photocathode to form a tandem device for unbiased water splitting, the photoanode is operating at around 0.6 V_{RHE} (*Science*, **2014**, 343, 990-994; *Nat. Energy*, **2016**, 2, 16191). Therefore, the PEC activity and stability of the FeOOH/BVO photoanode at 0.6 V_{RHE} are important for practical applications in the photoanode-photocathode configuration.

Thank you for your suggestions. We have added the data after 500 hours. As can be observed in Fig. 4g, gradual decay of the photocurrent can be observed after 520 h. According to the SEM image (Supplementary Fig. 29), some FeOOH particles disappeared and caused the direct exposure of the BiVO₄ particles to the electrolyte. Therefore, photocorrosion of BiVO₄ leads to the gradual decay of the photocurrent.

We also measured the stability at 1.23 V_{RHE}. As shown in Supplementary Fig. 30, our BVO-ΔO_v/FeOOH photoanode also shows excellent stability at 1.23 V_{RHE} for up to 470 h, and gradual decay can be observed after 470 h. Similarly, partial degradation of the surface FeOOH particles can also be observed (Supplementary Fig. 31).

Changes to the revised manuscript are shown below.

Main Manuscript (Results):

Page 21: Since the BVO-ΔO_v/FeOOH photoanode exhibits the highest ABPE value at around 0.6 V vs. RHE (Fig. 4f) and the photoanode is generally operated at around 0.6 V vs. RHE in a photoanode-photocathode tandem device, we measured the long-term stability performance of a BVO-ΔO_v/FeOOH film at 0.6 V vs. RHE.

Page 22: Gradual decay of the photocurrent density can be observed after around 520 h, and the photocurrent density is decreased to 3.6 mA cm⁻² at 540 h. According to the SEM image (Supplementary Fig. 29), some FeOOH particles disappear that lead to the direct exposure of the BiVO₄ particles to the electrolyte. Therefore, photocorrosion of BiVO₄ leads to the gradual decay of the photocurrent density. The BVO-ΔO_v/FeOOH film also shows excellent stability at 1.23 V vs. RHE for 470 h, and gradual decay of the photocurrent density can be observed in the range of 470-520 h (Supplementary Discussion and Supplementary Fig. 30). Similarly, the decay of photocurrent density is also attributed to the dissolution of FeOOH particles that lose the protection of BVO₄ particles (Supplementary Fig. 31).

Fig. 4g. J-t curve of the BVO-ΔO_v/FeOOH photoanode at 0.6 V vs. RHE under AM 1.5G illumination in a 1 M borate buffer electrolyte (pH 9.5).

Supplementary Information

Supplementary Fig. 29. Morphology characterization after stability test. SEM image of BVO-ΔO_v/FeOOH after 540 h of stability test at 0.6 V vs. RHE.

Page 34: The long-term stability performance of a BVO-ΔO_v/FeOOH film for PEC water splitting was also measured at 1.23 V vs. RHE under consecutive AM 1.5G illumination. As shown in Supplementary Fig. 30, more fluctuations of the photocurrent

densities can be observed compared to the measurement at 0.6 V vs. RHE, which is attributed to the evolution of much more oxygen bubbles from the photoanode. A stable photocurrent density of 6.8 mA cm^{-2} is observed until around 470 h. Gradual decay of the photocurrent density can be observed in the range of 470-520 h, decreasing to 4.9 mA cm^{-2} at 520 h.

Supplementary Fig. 30. Stability test at 1.23 V vs. RHE. J-t curve of the BVO-ΔO_v/FeOOH photoanode at 1.23 V vs. RHE under AM 1.5G illumination in a 1 M borate buffer electrolyte (pH 9.5).

Supplementary Fig. 31. Morphology characterization after stability test. SEM image of BVO-ΔO_v/FeOOH after 520 h of stability test at 1.23 V vs. RHE.

6) What is actual area of BiVO₄ and Si PV component in PEC-PV system for Figure 5e? Are both area kept same? Which configuration was used, tandem or parallel in all cases??

Similarly, what is the area of PEC-PV system in graph Figure 5g?

Explain all in caption and methodology.

Reply: Thanks a lot for your very helpful suggestions. We are sorry for ignoring this important information. The actual area of BiVO₄ and Si PV component in the PEC-PV system for Figure 5f (Figure 5e in the previous version) is 0.126 cm².

We make sure that both areas were kept the same. When measuring the J-V curve of the BiVO₄ photoanode, an opaque mask with an expose circle (diameter: 4 mm) was covered on the BiVO₄ photoanode, the other unexposed area was sealed by epoxy resin to avoid the contact with electrolyte. When measuring the J-V curve of the Si PV, the Si PV was covered by the BiVO₄ photoanode with an opaque mask to make sure that light can only be illuminated through the expose circle with a diameter of 4 mm (exposed area: 0.126 cm²).

The tandem configuration was used in all cases. Light was illuminated through the BiVO₄ photoanode first and then to the Si PV.

The I-t curve shown in Figure 5g is the PEC-PV device with dimensions of 3 cm × 3 cm (the exposed area is 6.25 cm²).

All the required information has been added in the caption and methodology.

Changes to the revised manuscript are shown below.

Main Manuscript (Results, Methods):

Page 24-25 (caption of Fig. 5): **Fig. 5. Unassisted solar water splitting with a BiVO₄-PV artificial leaf.** Scheme of a BVO-ΔO_v/FeOOH-PV artificial leaf with circuit connection (a) and charge transportation (b). c A wireless BVO-ΔO_v/FeOOH-PV artificial leaf with dimensions of 3 cm × 3 cm for water splitting under Xe lamp light (100 mW cm⁻²). d Digital image of a wireless BVO-ΔO_v/FeOOH-PV artificial leaf with dimensions of 21 cm × 21 cm (exposed area: 306.25 cm²). e A wireless BiVO₄-PV

artificial leaf with dimensions of 21 cm × 21 cm (exposed area: 306.25 cm²) for water splitting under natural sunlight. **f** J–V curves of a BVO-ΔO_v/FeOOH photoanode and PV behind the BVO-ΔO_v/FeOOH photoanode in tandem with an exposed area of 0.126 cm² under AM 1.5 G irradiation (100 mW cm⁻²). **g** Photocurrent density-time curve of a wired BVO-ΔO_v/FeOOH-PV artificial leaf with dimensions of 3 cm × 3 cm (exposed area: 6.25 cm²) under AM 1.5 G irradiation. **h** The corresponding H₂ and O₂ evolution of a of a wired BVO-ΔO_v/FeOOH-PV artificial leaf with dimensions of 3 cm × 3 cm (exposed area: 6.25 cm²) under AM 1.5 G irradiation. Comparison of **(i)** STH efficiency vs. area and **(j)** STH efficiency vs. operation time for BiVO₄-based PEC-PV devices, all perovskite PEC-PV devices, and other PEC-PV devices. (Detailed information is listed in Supplementary Table 9). Relative environmental impacts of the three scenarios for hydrogen production (normalized to the highest value among three scenarios for each impact category): **(k)** classified by resources and emissions, and **(l)** classified by unit process.

Page 35-36: H₂ and O₂ evolution measurements

As shown in Supplementary Figs. 38 and 40, H₂ and O₂ evolution performance of the wired and wireless BVO-ΔO_v/FeOOH–PV artificial leaves with dimensions up to 3 cm × 3 cm were measured in a photocatalytic activity evaluation system (Beijing China Education Au-Light Co., Ltd., CEL-PAEM-D8) connected with a gas chromatography (Beijing China Education Au-Light Co., Ltd., GC-7920). A Xe 300 W lamp (CEL-S300, CEAULIGHT) equipped with an AM 1.5 G filter was used as the light source. The light intensity at the artificial leaf was carefully calibrated to 100 mW cm⁻² with the spectrum matching the standard AM 1.5G spectrum. A 1 M potassium borate buffer (pH=9.5) was used as the electrolyte for all measurements. The produced gases were analyzed every 30 min or 1 h automatically controlled by the software.

Wireless artificial leaves with dimensions of 6 cm × 6 cm (exposed area: 25 cm²), 9 cm × 9 cm (exposed area: 56.25 cm²), 12 cm × 12 cm (exposed area: 100 cm²) and 21 cm × 21 cm (exposed area: 306.25 cm²) were sealed in home-made quartz reactors with dimensions of 15 cm × 15 cm × 10 cm and 30 cm × 30 cm × 10 cm (Supplementary Fig. 41), and the produced gas samples were taken out with a syringe every 30 min, and injected in GC every for testing.

7) Technical specification of Si PV devices are missing, include all information such as

J_{sc} , V_{oc} , FF, PCE of Si PV devices with small solar cells and large-modules. In Photograph, readers can not see Si PV Panel. Keep photographs of BiVO_4 and Si PV panel properly, how the both BiVO_4 and Si PV panel were arranged and connected in PEC-PV system?

Reply: Thank you very much for your valuable suggestions. All values of J_{sc} , V_{oc} , FF, PCE of Si PV devices have been provided, as shown in Supplementary Table 8.

The photographs of BiVO_4 and Si PV panel have been added in Supplementary Fig. 34. The arrangement and connection of BiVO_4 and Si PV have been illustrated in Supplementary Figs. 32-33.

The corresponding discussion has been added in the main text, as highlighted in Page 22-23 in the revised manuscript.

Changes to the revised manuscript are shown below.

Main Manuscript (Results):

Page 22-23: In this study, standalone artificial leaves with dimensions of 1 cm × 1 cm, 3 cm × 3 cm, 6 cm × 6 cm, 9 cm × 9 cm, 12 cm × 12 cm, and 21 cm × 21 cm were fabricated by integrating with a silicon cell (Supplementary Figs. 32-35). The J-V curves of a Si PV with and without the surface covered by a $\text{BVO-}\Delta\text{O}_v/\text{FeOOH}$ photoanode are shown in Supplementary Fig. 36. Detailed information of the J_{sc} , V_{oc} , FF, and PCE of the Si PV panels is summarized in Supplementary Table 8. The circuit connection mechanism and the charge transfer properties of the artificial leaf are shown in Fig. 5a, b. Detailed connections between the $\text{BVO-}\Delta\text{O}_v/\text{FeOOH}$ photoanodes and the Si PV panel are shown in Supplementary Figs. 33, 34.

Supplementary Information:

Supplementary Fig. 33. Scheme of the assembly of a 7 × 7 array artificial leaf. Illustration of a 21 cm × 21 cm Si PV panel, a BVO-ΔO_v/FeOOH photoanode (21 cm × 21 cm), and different views of the artificial leaf.

Supplementary Fig. 34. The structure of a 2 × 2 array artificial leaf. Digital images of a 2 × 2 (6 cm × 6 cm) array artificial leaf with different components.

Supplementary Fig. 36. Performance of Si PVs. J-V curves of Si PVs with and without the surface covered by a BVO- ΔO_v /FeOOH photoanode.

Supplementary Table 8. J-V Parameters of the Si PV panels.

Parameter	Value
J_{sc}	16 mA cm ⁻²
V_{oc}	1.5 V
FF	0.75
PCE	18%

8) What is the gridlines indicate in Figure 5a, 5d?

Also Figure 5 rearrangement is necessary as per follow,

Old Figure 5b (both schematics) \diamond Figure 5a

Old Figure 5c \diamond Figure 5b

Old Figure 5a \diamond Figure 5c

Old Figure 5g \diamond Figure 5f

Old Figure 5h \diamond Figure 5g

Old Figure 5f ◊ Figure 5h

Reply: Thank you for your excellent suggestions. The gridlines are the conductive graphite tapes used to connect each BiVO₄ films. The large area BiVO₄ photoanodes were assembled by small BiVO₄ films with dimensions of 3 cm × 3 cm. Silver conductive paint and conductive graphite tapes were used to connect each small BiVO₄ films to ensure the electric conductivity.

Figure 5 has been rearranged according to your suggestions. We attached the revised figure below for your information.

Fig. 5. Unassisted solar water splitting with a BiVO₄-PV artificial leaf. Scheme of a BVO-ΔO_v/FeOOH-PV artificial leaf with circuit connection (a) and charge transportation (b). c A wireless BVO-ΔO_v/FeOOH-PV artificial leaf with dimensions of 3 cm × 3 cm for water splitting under Xe lamp light (100 mW cm⁻²). d Digital image

of a wireless BVO- $\Delta\text{O}_v/\text{FeOOH}$ -PV artificial leaf with dimensions of 21 cm \times 21 cm (exposed area: 306.25 cm²). **e** A wireless BiVO₄-PV artificial leaf with dimensions of 21 cm \times 21 cm (exposed area: 306.25 cm²) for water splitting under natural sunlight. **f** J–V curves of a BVO- $\Delta\text{O}_v/\text{FeOOH}$ photoanode and PV behind the BVO- $\Delta\text{O}_v/\text{FeOOH}$ photoanode in tandem with an exposed area of 0.126 cm² under AM 1.5 G irradiation (100 mW cm⁻²). **g** Photocurrent density-time curve of a wired BVO- $\Delta\text{O}_v/\text{FeOOH}$ -PV artificial leaf with dimensions of 3 cm \times 3 cm (exposed area: 6.25 cm²) under AM 1.5 G irradiation. **h** The corresponding H₂ and O₂ evolution of a of a wired BVO- $\Delta\text{O}_v/\text{FeOOH}$ -PV artificial leaf with dimensions of 3 cm \times 3 cm (exposed area: 6.25 cm²) under AM 1.5 G irradiation. Comparison of **(i)** STH efficiency vs. area and **(j)** STH efficiency vs. operation time for BiVO₄-based PEC-PV devices, all perovskite PEC-PV devices, and other PEC-PV devices. (Detailed information is listed in Supplementary Table 9). Relative environmental impacts of the three scenarios for hydrogen production (normalized to the highest value among three scenarios for each impact category): **(k)** classified by resources and emissions, and **(l)** classified by unit process.

Supplementary Information

Supplementary Fig. 32. Scheme of the assembly of large-area photoanodes. a An example of assembling a 2 \times 2 array BVO- $\Delta\text{O}_v/\text{FeOOH}$ photoanode (6 cm \times 6 cm). **b** schemes of 3 \times 3 (9 cm \times 9 cm), 4 \times 4 (12 cm \times 12 cm), and 7 \times 7 (21 cm \times 21 cm) array BVO- $\Delta\text{O}_v/\text{FeOOH}$ photoanodes.

9) Considering the title of the manuscript, author should add the studying the geometrical area effect of PEC devices. i.e., J-V performance of different size (0.126 cm², 1 cm², 9 cm², 36 cm², 100 cm² and 440 cm²) of BiVO₄ photoanodes.

Reply: Thank you for your professional suggestions. The J-V performance of BiVO₄ photoanodes with different sizes of 1 cm² (exposed area: 0.126 cm²), 1 cm² (exposed area: 1 cm²), 9 cm² (exposed area: 6.25 cm²), 36 cm² (exposed area: 25 cm²), 81 cm² (exposed area: 100 cm²), 144 cm² (exposed area: 100 cm²) and 440 cm² (exposed area: 56.25 cm²) have been added, as shown in Supplementary Fig. 25. We attached the figure below for your information.

Supplementary Fig. 25. Analysis of the geometrical area effect on the photocurrent density. Photocurrent density versus potential curves of BVO- Δ O_v/FeOOH photoanodes with different areas in a 1 M borate buffer electrolyte 1 M Na₂SO₃ (pH 9.5) under AM 1.5G illumination.

Changes to the revised manuscript are shown below.

Main Manuscript (Results):

Page 20-21: For possible scale-up applications, the geometrical area effect of the BVO- Δ O_v/FeOOH photoanode on the photocurrent density was systematically studied. As demonstrated in Supplementary Fig. 25, the photocurrent density decreases with the increase area of the BVO- Δ O_v/FeOOH photoanode.

10) STH efficiency of 8.8 % of which size of PEC-PV system?

Discuss STH efficiency of above varying the size PEC-PV systems and gas evolution rate for each.

Effect of area on STH efficiency performance of PEC-PV system, generally STH efficiency decrease while increasing the area.

3 x 3 (9 cm²) PEC-PV system

6 x 6 (36 cm²) PEC-PV system

10 x 10 (90 cm²) PEC-PV system

21 x 21 (440 cm²) PEC-PV system

Author should can add BiVO₄-Si PV cell as wired or wireless artificial device. Wired device can help to measured J-V curve + stability + gas productivity, while wireless artificial device can only provide gas productivity. So author should compare in the way to provide useful information systematically. Figure 5 neither clarify nor represent anything.

Reply: Thanks a lot for your constructive suggestions to further improve the quality of our manuscript. The STH efficiency of 8.8% we achieved is based on the PEC-PV system with a size of 0.126 cm². When the area increases, the resistance within the film also increases, and the resistance for mass transfer between the electrode and the electrolyte also increases, which lead to the decrease of the STH efficiency. The STH efficiencies of the PEC-PV systems with different sizes of 3 × 3 (9 cm²), 6 × 6 (36 cm²), 10 × 10 (90 cm²) and 21 × 21 (441 cm²) were measured, as shown in Supplementary Fig. 37.

We have measured the performances of the wired and wireless BiVO₄-Si artificial leaf with size of 3 cm × 3 cm as an example. As shown in Fig. 5h and Supplementary Fig. 42, the gas evolution performances are very similar. In fact, in our design, the wireless device also has a short wire to connect the BiVO₄ photoanode and the Si PV, and the short wire was buried in a water-proof silicone coating to avoid the contact with electrolyte. Therefore, the working mechanism and charge transport properties in the wired and wireless devices are the same. The only difference is that we can use the electrochemical workstation to measure the J-V and J-t curves with the wired device, while the wireless device is an integrated device that cannot be connected with electrochemical workstation.

We are sorry for ignoring the necessary information in the caption of Figure 5. All required information has been added in the revised manuscript, as highlighted in Page 24-25.

Changes to the revised manuscript are shown below.

Main Manuscript (Results):

Page 23: Based on the operating photocurrent densities, the STH efficiencies of other artificial leaves with different exposed area of 1, 6.25, 25, 56.25, 100, and 306 cm² are 8.6%, 7.9%, 7.1%, 6.1%, 5.0%, and 3.5%, respectively (Supplementary Fig. 37).

Page 24-25: Fig. 5. Unassisted solar water splitting with a BiVO₄-PV artificial leaf. Scheme of a BVO-ΔO_v/FeOOH-PV artificial leaf with circuit connection (a) and charge transportation (b). c A wireless BVO-ΔO_v/FeOOH-PV artificial leaf with dimensions of 3 cm × 3 cm for water splitting under Xe lamp light (100 mW cm⁻²). d Digital image of a wireless BVO-ΔO_v/FeOOH-PV artificial leaf with dimensions of 21 cm × 21 cm (exposed area: 306.25 cm²). e A wireless BiVO₄-PV artificial leaf with dimensions of 21 cm × 21 cm (exposed area: 306.25 cm²) for water splitting under natural sunlight. f J–V curves of a BVO-ΔO_v/FeOOH photoanode and PV behind the BVO-ΔO_v/FeOOH photoanode in tandem with an exposed area of 0.126 cm² under AM 1.5 G irradiation (100 mW cm⁻²). g Photocurrent density-time curve of a wired BVO-ΔO_v/FeOOH-PV artificial leaf with dimensions of 3 cm × 3 cm (exposed area: 6.25 cm²) under AM 1.5 G irradiation. h The corresponding H₂ and O₂ evolution of a of a wired BVO-ΔO_v/FeOOH-PV artificial leaf with dimensions of 3 cm × 3 cm (exposed area: 6.25 cm²) under AM 1.5 G irradiation. Comparison of (i) STH efficiency vs. area and (j) STH efficiency vs. operation time for BiVO₄-based PEC-PV devices, all perovskite PEC-PV devices, and other PEC-PV devices. (Detailed information is listed in Supplementary Table 9). Relative environmental impacts of the three scenarios for hydrogen production (normalized to the highest value among three scenarios for each impact category): (k) classified by resources and emissions, and (l) classified by unit process.

Page 26-27: Gas evolution performances of wireless artificial leaves with dimensions up to 3 cm × 3 cm were measured in a sealed reactor connecting to a GC (Supplementary Fig. 40). Larger area artificial leaves with dimensions of 6 cm × 6 cm

(exposed area: 25 cm²), 9 cm × 9 cm (exposed area: 56.25 cm²), 12 cm × 12 cm (exposed area: 100 cm²) and 21 cm × 21 cm were sealed in home-made quartz reactors with dimensions of 15 cm × 15 cm × 10 cm and 30 cm × 30 cm × 10 cm (Supplementary Fig. 41), and the produced gas samples were taken out with a syringe, and injected in GC every hour for testing. The gas production performance 3 wireless artificial leaves were measured, and the average H₂ and O₂ evolution performances with error bars are shown in Supplementary Fig. 42. It can be observed that the performance of the wireless artificial leaf is similar to that of the wired artificial leaf (Fig. 5h). Based on the gas evolution performances (Supplementary Figs. 42, 43) and Supplementary Equation 19, the STH efficiencies of the artificial leaves with exposed areas of 0.126 cm², 1 cm², 6.25 cm², 25 cm², 56.25 cm², 100 cm² and 306.25 cm² can be calculated as 8.4%, 8.2%, 7.2%, 6.3%, 5.0%, 4.2%, and 2.7% (Supplementary Fig. 44). It can be observed that the STH efficiencies calculated from the production of hydrogen are slightly lower than those calculated from the operating photocurrent densities, which is because the Faradaic efficiency for PEC water splitting is not 100%.

Supplementary Fig. 37. STH efficiencies of BVO- Δ Ov/FeOOH-PV artificial leaves with different sizes. J-V curves of a BVO- Δ Ov/FeOOH photoanode and PV behind the BVO- Δ Ov/FeOOH photoanode in tandem with different exposed areas of 0.126, 1, 6.25, 25, 56.25, 100, and 306.25 cm² under AM 1.5 G irradiation.

Supplementary Fig. 42. Gas evolution performance for a wireless artificial leaf. Average H₂ and O₂ evolution performances of 3 artificial leaves with dimensions 3 cm × 3 cm (exposed area 6.25 cm²). Error bars are included.

Supplementary Fig. 43. Gas evolution performance for wireless artificial leaves with different sizes. **(a)** 1 cm × 1 cm (control exposed area 0.126 cm²), **(b)** 1 cm × 1 cm (exposed area: 1 cm²), **(c)** 6 cm × 6 cm (exposed area: 25 cm²), **(d)** 9 cm × 9 cm (exposed area: 56.25 cm²), **(e)** 12 cm × 12 cm (exposed area: 100 cm²) and **(f)** 21 cm × 21 cm (exposed area: 306.25 cm²).

Supplementary Fig. 44. STH efficiencies for wireless artificial leaves with different sizes. STH efficiencies of wireless artificial leaves with different sizes calculated based on their hydrogen evolution performances: 1 cm × 1 cm (control exposed area 0.126 cm²), 1 cm × 1 cm (exposed area: 1 cm²), 3 cm × 3 cm (exposed area 6.25 cm²), 6 cm × 6 cm (exposed area: 25 cm²), 9 cm × 9 cm (exposed area: 56.25 cm²), 12 cm × 12 cm (exposed area: 100 cm²) and 21 cm × 21 cm (exposed area: 306.25 cm²).

11) How the gas productivity was measured of artificial leaf, did author used specific PEC reactor with membrane, please provide the data of PEC cell reactor.

Reply: Thank you for your valuable comments. We measured the gas productivity of the wireless artificial leaf in a photocatalytic activity evaluation system connected with a gas chromatography, as shown in Supplementary Fig. 40. Before measuring the gas productivity, the system was vacuumed to remove all gases. A Xe light equipped with an AM 1.5 G filter was used as the light source, and the light intensity illuminated at the artificial leaf was carefully calibrated to 100 mW cm⁻². The produced gas was analyzed every 30 min or 1 h. For large area artificial leaves, the artificial leaf was placed in a home-made quartz reactor, which is then sealed in a glove-box filled with Ar. After light illumination for 30 min or 1 h, the produced gases were taken out by a

syringe and inject into the gas chromatography for analysis.

The gas productivity of the wired artificial leaf is measured in a sealed reactor, as shown in Supplementary Fig. 38. The electrodes were connected in the electrochemical workstation to measure the photocurrent. Before measuring the gas productivity, the system was vacuumed to remove all gases. A Xe light equipped with an AM 1.5 G filter was used as the light source, and the light intensity illuminated at the artificial leaf was carefully calibrated to 100 mW cm^{-2} . The produced gas was analyzed every 30 min (or 1 h).

Changes to the revised manuscript are shown below.

Main Manuscript (Results):

Page 25: The BVO- ΔO_v /FeOOH-PV artificial leaf with dimensions of $3 \text{ cm} \times 3 \text{ cm}$ was placed in a sealed reactor connecting to an electrochemical workstation to monitor the photocurrent densities (Supplementary Fig. 38). The BVO- ΔO_v /FeOOH-PV artificial leaf can achieve water splitting for 50 h with a photocurrent density retention rate of 92% (Fig. 5g).

Page 26: The system was vacuumed every 10 h to remove all gases for one cycle test. As shown in Fig. 5h, the generated H_2/O_2 show no noticeable decrease, with production rates of $1117/551 \mu\text{mol cm}^{-2}$ in 10 h of consecutive illumination, with a ratio of round 2:1. This confirms the long-term operational capability of the fabricated artificial leaf.

Gas evolution performances of wireless artificial leaves with dimensions up to $3 \text{ cm} \times 3 \text{ cm}$ were measured in a sealed reactor connecting to a GC (Supplementary Fig. 40). Larger area artificial leaves with dimensions of $6 \text{ cm} \times 6 \text{ cm}$ (exposed area: 25 cm^2), $9 \text{ cm} \times 9 \text{ cm}$ (exposed area: 56.25 cm^2), $12 \text{ cm} \times 12 \text{ cm}$ (exposed area: 100 cm^2) and $21 \text{ cm} \times 21 \text{ cm}$ were sealed in home-made quartz reactors with dimensions of $15 \text{ cm} \times 15 \text{ cm} \times 10 \text{ cm}$ and $30 \text{ cm} \times 30 \text{ cm} \times 10 \text{ cm}$ (Supplementary Fig. 41), and the produced gas samples were taken out with a syringe every 30 min, and injected in GC for testing.

Supplementary Fig. 38. Gas evolution measurement for a wired artificial leaf. **a** Digital photo of gas evolution measurement for a wired artificial leaf connected with a gas chromatography. **b** Digital photo of a reactor for measuring the gas evolution and photocurrent densities of a wired artificial leaf. The reactor can measure gas evolution of a wired artificial leaf with dimensions up to 3 cm × 3 cm.

Supplementary Fig. 40. Gas evolution measurement for a wireless artificial leaf. Digital photo of gas evolution measurement for a wireless artificial leaf in a photocatalytic activity evaluation system connected with a gas chromatography. The reactor can measure gas evolution of a wireless artificial leaf with dimensions up to 3 cm × 3 cm.

Supplementary Fig. 41. Gas evolution measurement for a wireless artificial leaf. Digital photos of (a) a sealed quartz reactor with dimensions of 15 cm × 15 cm × 10 cm (inside is a wireless artificial leaf with dimensions of 9 cm × 9 cm as reference), and (b) a sealed quartz reactor with dimensions of 30 cm × 30 cm × 10 cm.

12) Is this 50 hours a maximum period of time measured for PEC-PV artificial leaf? If device was degraded even after then put the data and explain the reason of degradation beyond 50h.

Reply: Many thanks for your valuable suggestions. The BVO- $\Delta\text{O}_v/\text{FeOOH}$ -PV artificial leaf can achieve water splitting for 50 h with a photocurrent density retention rate of 92% (Fig. 5g). It should be mentioned that the encapsulation in the circuit of the artificial leaf to avoid the direct contact of the electrolyte is very important to achieve long-term stability. We found that if the encapsulation is not well, the photoelectrochemical corrosion of the circuit causes the fluctuation of the output photocurrent densities (Supplementary Fig. 39). Since the BVO- $\Delta\text{O}_v/\text{FeOOH}$ photoanode can continuously achieve PEC water splitting for over 520 h (Fig. 4g), the BVO- $\Delta\text{O}_v/\text{FeOOH}$ -PV artificial leaf is also expected to achieve hundreds hours of stability if the PV and connected circuit are encapsulated well.

Changes to the revised manuscript are shown below.

Main Manuscript (Results):

Page 25: The BVO- $\Delta\text{O}_v/\text{FeOOH}$ -PV artificial leaf can achieve water splitting for 50 h with a photocurrent density retention rate of 92% (Fig. 5g). It should be mentioned that the encapsulation in the circuit of the artificial leaf to avoid the direct contact of the electrolyte is very important to achieve long-term stability. We found that if the encapsulation is not well, the photoelectrochemical corrosion of the circuit causes the

fluctuation of the output photocurrent densities (Supplementary Fig. 39).

Page 26: Since the BVO- $\Delta\text{O}_v/\text{FeOOH}$ photoanode can continuously achieve PEC water splitting for over 520 h (Fig. 4g), the BVO- $\Delta\text{O}_v/\text{FeOOH}$ -PV artificial leaf is also expected to achieve hundreds hours of stability if the PV and connected circuit are encapsulated well.

Supplementary Fig. 39. Stability measurement of a wired artificial leaf with imperfect encapsulation of the connections. J-V curve of a wired BVO- $\Delta\text{O}_v/\text{FeOOH}$ -PV wired artificial leaf with dimensions of 3 cm \times 3 cm.

13) For LCA study, author should state couple sentence about global warming potential (GWP), as new green hydrogen production technology should not affect the global warming much compared to existing technologies such steam methane reforming (SMR) and other green hydrogen production technology. Refer paper: Chem. Soc. Rev., 53, 2388-2434, 2024.

Reply: Many thanks for your valuable suggestions. We totally agree that it is important to state the global warming potential of our PEC water splitting technology. We have carefully read the excellent paper, which is very helpful. Additional discussion has been added, as highlighted in Page 28 in the revised manuscript. The paper has been cited as Ref. 63.

Changes to the revised manuscript are shown below.

Main Manuscript (Results):

Page 28: The global warming potential (GWP, $\text{kg}_{\text{CO}_2 \text{ eq.}} \text{kg}_{\text{H}_2}^{-1}$) of a hydrogen production technology is also important to assess the environmental impacts⁶³. The GWP values of natural gas reforming (Scenario 1), electrocatalytic water splitting (Scenario 2), and photoelectrochemical water splitting (Scenario 3) are 141.3, 8.7, and 2.1 $\text{kg}_{\text{CO}_2 \text{ eq.}} \text{kg}_{\text{H}_2}^{-1}$, respectively.

References:

63 Vilanova, A., Dias, P., Lopes, T. & Mendes, A. The route for commercial photoelectrochemical water splitting: a review of large-area devices and key upscaling challenges. *Chem. Soc. Rev.* 53, 2388-2434 (2024).

14) In the attempt of scalability and LCA of PEC system, author should briefly discuss the techno-economic part. e.g., Cost of Photoanode device and PEC system.

Reply: Thanks a lot for your excellent suggestions. Yes, the techno-economic part is also very important for scalability. We have added the techno-economic analysis of our developed PEC-PV system, natural gas reforming and electrocatalytic water splitting for hydrogen production. As shown in Supplementary Fig. 45, the cost of PEC-PV system is a higher than natural gas reforming, while lower than electrocatalytic water splitting. Additional discussion has been added, as highlighted in Page 28-29 in the revised manuscript.

Changes to the revised manuscript are shown below.

Main Manuscript (Results):

Page 28-29: In addition, the cost of producing 1 kg of H₂ using the above three technologies were also compared. As shown in Supplementary Fig. 45, the cost of photoelectrochemical water splitting is higher than natural gas reforming while lower than electrocatalytic water splitting. However, if taking the carbon tax into account, photoelectrochemical water splitting is competitive for hydrogen production in the future market.

Supplementary Fig. 45. Techno-economic comparison of different technologies for hydrogen production. Comparison for the cost of producing 1 kg of hydrogen from three different technologies: natural gas reforming (Scenario 1), electrocatalytic water splitting (Scenario 2), and photoelectrochemical water splitting (Scenario 3).

15) Author should cite the scalable PEC papers (Nat. Energy 9, 272–284, 2024; Energy Environ. Sci., 17, 3604-3617, 2024; J. Power Sources, 398, 224, 2018; J. Power Sources, 454, 2020; Sustainable Energy Fuels, 8, 3726-3739; 2024; Small Methods, 7, 10, 2300619, 2023) in main script and compare the STH efficiency in table in Supplementary information.

Reply: We appreciate your precious suggestions. We are sorry for ignoring all these highly related papers. All these papers have been discussed and cited. A table (Supplementary Table 9) has been added in the Supplementary information to compare the STH efficiencies of different systems. All these excellent papers have been cited as Ref. 10, 56-60, as highlighted in Page 39, and 45.

Changes to the revised manuscript are shown below.

Main Manuscript (Results):

Page 27: As shown in Fig. 5i, the STH efficiencies of our developed BVO- $\Delta\text{O}_v/\text{FeOOH}$ -PV artificial leaves are the best amongst all BiVO_4 -PV based unbiased water splitting systems^{10,30,39-60}. The stability of our artificial leaf is also comparable to other state-of-the-art devices(Fig. 5j, Supplementary Table 9).

Fig. 5i Comparison of STH efficiency vs. area for BiVO_4 -based PEC-PV devices, all perovskite PEC-PV devices, and other PEC-PV devices. **j** Comparison of STH efficiency vs. operation time for BiVO_4 -based PEC-PV devices, all perovskite PEC-PV devices, and other PEC-PV devices (Detailed information is listed in Supplementary Table 9).

References:

- 10 Hansora, D. *et al.* All-perovskite-based unassisted photoelectrochemical water splitting system for efficient, stable and scalable solar hydrogen production. *Nat. Energy* **9**, 272-284 (2024).
- 56 Jeong, W. *et al.* Large-area all-perovskite-based coplanar photoelectrodes for scaled-up solar hydrogen production. *Energy Environ. Sci.* **17**, 3604-3617 (2024).
- 57 Vilanova, A., Lopes, T. & Mendes, A. Large-area photoelectrochemical water splitting using a multi-photoelectrode approach. *J. Power Sources* **398**, 224-232 (2018).
- 58 Vilanova, A. *et al.* Solar water splitting under natural concentrated sunlight using a 200 cm² photoelectrochemical-photovoltaic device. *J. Power Sources* **454**, 227890 (2020).
- 59 Maragno, A. R. A. *et al.* Thermally integrated photoelectrochemical devices with perovskite/silicon tandem solar cells: a modular approach for scalable direct water splitting. *Sustainable Energy Fuels* **8**, 3726-3739 (2024).
- 60 Xu, Z., Chen, L., Brabec, C. J. & Guo, F. All Printed Photoanode/Photovoltaic Mini-Module for Water Splitting. *Small Methods* **7**, 2300619 (2023).

Reviewer #2 (Remarks to the Author):

In the present manuscript the authors study a BiVO_4 photoanode to fabricate an artificial leaf coupled with a Si-based PV cell. Along this study, the authors performed a detailed structural, morphological, chemical and photoelectrochemical characterization of the employed materials through a wide range of different techniques. Additionally, a scaled-up artificial leaf has been successfully developed and tested under relevant working conditions. The manuscript is well written and the experiments well performed. However, even the manuscript is of the true interest of the solar fuels community and it is well aligned with the scope of Nature Communications, there are some major points that have to be addressed before publication in order to achieve the high-quality standards required in this prestigious journal.

Reply: We are very grateful for your affirmation of our work, and your guidance to further improve the quality of our work. We have carefully considered all of your suggestions and comments. Please read our point-to-point response and revision of our work as below.

1. The quality and contrast of the SEM images shown in Figure 1 need to be improved.

Reply: Thanks for your suggestion. The brightness and contrast of all SEM images in Figure 1 have been modified. Here we attached the modified SEM images in Figure 1 for your information.

Fig. 1. SEM images of (a) BSO-3, (b) BSO-5 and (c) BSO-7. Insets: local

magnification of the BSO-*x* samples. SEM images of (e) BVO- ΔO_v -3, (f) BVO- ΔO_v -5 and (g) BVO- ΔO_v -7. Insets: the cross-sectional SEM images of the BVO- ΔO_v -*x* samples.

2. The authors need to explain better what is the difference between BVO-5 and BVO-5a. It is not clear for the reader how it is written leading to confusion between thermal treatments and electrolyte modifications (between nitric and acetic acid).

Reply: Thank you very much for your valuable suggestions. We are sorry for making the confusion. According to our characterizations, BVO-5 exhibits gradient distribution of oxygen vacancies within the film, while BVO-5a only have a small number of oxygen vacancies in the top surface.

To avoid any misunderstanding, we have renamed BVO-5 as BVO- ΔO_v and BVO-5a as BVO, respectively.

All the changes have been highlighted in the revised Manuscript.

Changes to the revised manuscript are shown below.

Main Manuscript (Introduction):

Page 4: Here, we develop an electrolyte recipe for the preparation of $\text{Bi}_{34.7}\text{O}_{36}(\text{SO}_4)_{16}$ precursor films, followed by solid thermal reaction with vanadyl acetylacetonate ($\text{VO}(\text{acac})_2$) to construct wormlike BiVO_4 photoanodes with a gradient distribution of oxygen vacancies (denoted as BVO- ΔO_v).

3. The authors have to report the photocurrent densities, IPCE and the rest of the PEC characterisation without hole scavenger to ease the comparison and benchmark of this work with other previous works on the state-of-the-art. Basically, Figure 2 has to be repeated in bare aqueous electrolyte without hole scavenger. On this regard, why the authors employed borate buffer instead of KPi buffer? Usually, charge separation efficiency versus applied potential (Figure 2c) is represented in continuous line.

Reply: Thanks a lot for your valuable suggestions. In Figure 2, we would like to exclude the effect of surface charge transfer properties, and focused on study the bulk

charge separation properties. Since no oxygen evolution cocatalyst was used, the photocurrent densities and IPCE performances of the photoanodes without hole scavenger should be very low. In this case, we are not clear which factors really affect the performance. Testing the PEC performance of the photoanode in the presence of hole scavenger has been widely accepted to study the bulk charge separation properties of photoanodes (*Adv. Energy Mater.*, **2016**, *6*, 1501645; *Science*, **2014**, *343*, 990-994).

The photocurrent densities and IPCE curves of the two photoanodes in the electrolyte without hole scavenger were also measured, as shown in Supplementary Fig. 9. The photocurrent densities of BVO- ΔO_v and BVO are 4.15 and 3.0 mA cm⁻² at 1.23 V vs. RHE. The IPCE values are 62 and 47%, respectively.

In a previous study, researchers confirmed that BiVO₄ photoanodes are more stable in borate buffer than KPi buffer (*Nat. Energy*, **2016**, *2*, 16191; *Nat. Commun.*, **2016**, *7*, 12012). Therefore, many works studying BiVO₄ photoanodes also used borate buffer (*Nat. Energy*, **2018**, *3*, 53-60; *Angew. Chem. Int. Ed.*, **2020**, *59*, 6213-6218; *Nat. Commun.*, **2024**, *15*, 9127).

Yes, charge separation efficiency versus applied potential curves should be in continuous lines. We are sorry that when output the figure we chose the wrong mode that only exhibited scatter dots. We have revised this figure, and we attached the figure below for your information.

Fig. 2c η_{sep} curves of BVO- ΔO_v and BVO.

Changes to the revised manuscript are shown below.

Main Manuscript (Results):

Page 8: Bulk charge separation efficiencies of the BVO- ΔO_v - x photoanodes were measured in a three-electrode cell with 0.2 M Na_2SO_3 as the hole scavenger under AM 1.5G illumination (100 mW cm^{-2}). Owing to the low activation energy and fast kinetics for the oxidation of SO_3^{2-} ions²⁷, all photogenerated holes reaching the surface of BVO- ΔO_v - x would be immediately consumed in the presence of SO_3^{2-} ions, and thus the effect of surface charge recombination on the photocurrent densities can be excluded.

Page 10: In the absence of Na_2SO_3 , the photocurrent densities and IPCE values of both BVO- ΔO_v and BVO decrease (Supplementary Discussion and Supplementary Fig. 9) due to the sluggish kinetics for OER.

Page 18: Since phosphate buffer electrolytes can slowly dissolve BiVO_4 and Ni-based catalysts^{20,38}, PEC water splitting performance of all samples were measured in a 1 M borate buffer without sacrificial agent.

References:

27 Seabold, J. A. & Choi, K.-S. Efficient and Stable Photo-Oxidation of Water by a Bismuth Vanadate Photoanode Coupled with an Iron Oxyhydroxide Oxygen Evolution Catalyst. *J. Am. Chem. Soc.* **134**, 2186-2192 (2012).

20 Kuang, Y. *et al.* Ultrastable low-bias water splitting photoanodes via photocorrosion inhibition and in situ catalyst regeneration. *Nat. Energy* **2**, 16191 (2016).

38 Toma, F. M. *et al.* Mechanistic insights into chemical and photochemical transformations of bismuth vanadate photoanodes. *Nat. Commun.* **7**, 12012 (2016).

Supplementary Information:

Page 16: In the absence of Na_2SO_3 , the photocurrent densities of both BVO- ΔO_v and BVO are much lower due to the sluggish OER kinetics on the BiVO_4 surfaces. As shown in Supplementary Fig. 9a, the photocurrent densities of BVO- ΔO_v and BVO are 4.15 and 3.0 mA cm^{-2} at 1.23 V vs. RHE under AM 1.5G illumination, respectively. BVO- ΔO_v exhibits an IPCE value of 62% in the wavelength range of 350-450 nm, while that of its BVO counterpart is 47% (Supplementary Fig. 9b).

Supplementary Fig. 9. Photoelectrochemical performance of the samples in the absence of Na₂SO₃. a Photocurrent density versus potential curves and **(b)** IPCE curves at 1.23 V vs. RHE of BVO-ΔO_v and BVO in a 1 M borate buffer electrolyte (pH 9.5) under AM 1.5G illumination.

4. Why the authors did not check the Vanadium edge on XAS instead of the Bi one? The V is the more sensitive edge to oxygen vacancies. Please take a look to the following reference and add to the manuscript: <https://doi.org/10.1021/acsami.2c07451>.

Reply: Many thanks for your constructive suggestions. We have done the vanadium edge XAS of two samples, as shown in Supplementary Fig. 15. Addition discussion has been added, as highlighted in Page 21 in Supplementary Information. The vanadium edge on XAS also support the presence of oxygen vacancies. The excellent paper has been cited as Ref. 36 to support our discussion.

Changes to the revised manuscript are shown below.

Main Manuscript (Results):

Page 16: XANES spectra at V *K*-edge of BVO-ΔO_v and BVO also confirm the presence of more oxygen vacancies in BVO-ΔO_v (Supplementary Discussion, Supplementary Fig. 15, and Supplementary Table 7)³⁶.

References:

36 Barawi, M. *et al.* Laser-Reduced BiVO₄ for Enhanced Photoelectrochemical Water Splitting. *ACS Appl. Mater. Interfaces* **14**, 33200-33210 (2022).

Supplementary Information:

Page 21-22: Supplementary Fig. 15a shows the XANES spectra at V *K*-edge of BVO- ΔO_v , BVO, V foil, V_2O_3 , and V_2O_5 . An intense pre-edge peak located at 5469 eV can be observed in all samples, which is associated with $1s-3d$ transitions. The edge positions of both BVO- ΔO_v and BVO are almost overlapped, which is slightly negative shifted compared to that of V_2O_5 , indicating the presence of low-valent V ions along with the V^{5+} ions in the samples. V *K*-edge radial distance $\chi(R)$ space spectra of BVO- ΔO_v and BVO were performed to obtain more information about the local structure around V ions (Supplementary Figs. 15b-d). As shown in Supplementary Fig. 15b, the peaks located at around 1.32 Å assigned to the V–O bond are observed in both BVO- ΔO_v and BVO samples, and no obvious peak shift can be observed. However, the slightly weaker intensity of the V–O peak in BVO- ΔO_v indicates the generation of more oxygen vacancies that leads to less amount of V–O bonds. The good fitting results of $\chi(R)$ and $\chi(k)$ space spectra (Supplementary Figs. 15c, d) with reasonable R-factors and the obtained fitting parameters (Supplementary Table 7) provide a quantitative of the V–O1 and V–O2 bonds. The shorter distance of the V–O2 bond in BVO- ΔO_v suggests the presence of more oxygen vacancies.

Supplementary Fig. 15. XANES spectra at V K-edge of the samples for oxygen vacancy analysis. **(a)** Normalized V K-edge XANES $\mu(E)$ spectra, and **(b)** V K-edge radial distance $\chi(R)$ space spectra of BVO- ΔO_v and BVO. Fourier-transformed (FT)-Extended X-ray absorption fine structure (EXAFS) fitting curves at R space of V K-edge of **(c)** BVO- ΔO_v and **(d)** BVO.

Supplementary Table 7. Fitting parameters of V K-edge EXAFS curve for BVO- ΔO_v and BVO.

Samples	Path ^a	R(\AA) ^b	N ^c	$\sigma^2(\text{\AA}^{-2})$ ^f	ΔE_0 (eV) ^g	Rf, %
BVO	V-O1	1.68 (1.72)	2 ^e	0.0006	12.36	0.09
	V-O2	2.29 (1.78)	2 ^e	0.0007		
	V-Bi	3.58 (3.58)	2 ^e	0.011		
BVO- ΔO_v	V-O1	1.68	2 ^e	0.0005	10.95	0.12
	V-O2	1.83	2 ^e	0.0005		
	V-Bi	3.57	2 ^e	0.012		

^a The distances for the path are from the crystal structure of BiVO₄ (1101208).

^b R: average distance between absorber and backscattered atoms.

^c N: coordination number.

^{d e} The coordination number is fixed according to the crystal structure model.

^f σ^2 : Debye-Waller factor.

^g ΔE_0 : the inner potential correction.

The data range used for fitting in R space (ΔR) is 1.0–5.0 \AA .

5. The authors should elaborate more in detail the structure and components of the artificial leaf. I strongly recommend the authors to include a detailed scheme of the different components.

Reply: Thanks a lot for your excellent suggestions. We are sorry that structure and components of the artificial leaf are not very clear in our previous version of the manuscript. We have added detailed photos and schemes to demonstrate the structure

and connection of the artificial leaf, as shown in Supplementary Fig. 32-35.

Changes to the revised manuscript are shown below.

Supplementary Information:

Page 36-38:

Supplementary Fig. 32. Scheme of the assembly of large-area photoanodes. a An example of assembling a 2×2 array BVO- Δ Ov/FeOOH photoanode. **b** schemes of 3×3 , 4×4 , and 7×7 array BVO- Δ Ov/FeOOH photoanodes.

Supplementary Fig. 33. Scheme of the assembly of a 7×7 array artificial leaf. Illustration of a $21 \text{ cm} \times 21 \text{ cm}$ Si PV panel, a BVO- Δ Ov/FeOOH photoanode ($21 \text{ cm} \times 21 \text{ cm}$), and different views of the artificial leaf.

Supplementary Fig. 34. The structure of a 2×2 array artificial leaf. Digital photos of a 2×2 array artificial leaf with different components.

Supplementary Fig. 35. Artificial leaves with different sizes. Digital photos of (a) a $21 \text{ cm} \times 21 \text{ cm}$ artificial leaf, and (b) artificial leaves with different dimensions of $1 \text{ cm} \times 1 \text{ cm}$, $3 \text{ cm} \times 3 \text{ cm}$, $6 \text{ cm} \times 6 \text{ cm}$, $9 \text{ cm} \times 9 \text{ cm}$.

Reviewer #3 (Remarks to the Author):

In this study, the authors present a compelling design for a BiVO_4 (BVO) photoanode characterized by a gradient distribution of oxygen vacancies, which generates strong dipole fields within the film to enhance charge separation. Notably, the device achieved a remarkable photocurrent density of approximately 7 mA/cm^2 at 1.23 V versus the

reversible hydrogen electrode (RHE), as well as exhibiting stability over a 500-hour duration. Additionally, by integrating the fabricated photoanode with a silicon solar cell in a tandem configuration, the authors achieved a solar-to-hydrogen (STH) conversion efficiency of 8.8%. Furthermore, they demonstrated the feasibility of scaling up the BVO photoanode and its integration with a silicon solar cell with an area of approximately 441 cm². The structural and performance evaluations are substantively supported by experimental data. However, several significant comments and suggestions must be addressed before the manuscript can be recommended for publication:

Reply: We are very grateful for your very positive evaluation of our work, and your guidance to further improve the quality of our work. We have carefully considered all of your suggestions and comments. Please read our point-to-point response and revision of our work as below.

1. The title of the manuscript requires revision. The authors demonstrated an STH conversion efficiency of approximately 8.8% using a small BVO photoanode with an area of 0.126 cm², rather than from a large-area BVO photoanode. It is therefore advisable for the authors to reconsider the title for clarity and accuracy.

Reply: We appreciate your wonderful suggestions. We have changed the title as “A standalone bismuth vanadate-silicon artificial leaf achieving 8.4% efficiency for zero-emission hydrogen production”. 8.8% efficiency in our previous title is calculated by the photocurrent density. As suggested by the reviewers, it is more accurate to calculate the efficiency by the hydrogen gas evolution. After carefully measuring the production of hydrogen gas, we have updated the efficiency to 8.4%.

2. Figure 2b: The reviewer recommends that the authors provide the estimated photocurrent for the photoanodes depicted in the graph, determined from the integrated IPCE spectra over the standard solar spectrum, to corroborate the photocurrent data presented in Figure 2a.

Reply: Many thanks for your suggestions. The estimated photocurrent densities integrated from IPCE spectra over the standard solar spectrum are shown in

Supplementary Fig. 8. It can be observed that the integrated photocurrent densities are 6.9 and 3.8 mA cm⁻², which are close to the photocurrent densities shown in Figure 2b. Therefore, the Xe light equipped with an AM 1.5 G filter is very close to the standard solar spectrum.

Changes to the revised manuscript are shown below.

Main Manuscript (Results):

Page 10: By integrating the IPCE curves of BVO- ΔO_v and BVO with the standard AM 1.5 G spectrum, the estimated photocurrent densities are 6.9 and 3.8 mA cm⁻², respectively (Supplementary Fig. 8), which are close to the measured photocurrent densities shown in Fig. 2a.

Supplementary Information:

Page 15:

Supplementary Fig. 8. Calculated photocurrent density curves of the samples. Photocurrent density curves of BVO- ΔO_v and BVO obtained by integrating their IPCE curves with standard AM 1.5G spectrum.

3. Page 18, lines 317-318: The authors should amend the phrase "To further evaluate the overall water splitting..." to "To further evaluate water splitting ...".

Reply: Thanks for your suggestions. We have changed the expression as "To further evaluate water splitting ...", as highlighted in Page 22 in the revised manuscript.

4. Figure 4h: The authors are encouraged to calculate the theoretical generation rates of O₂ and H₂ and to determine the Faradaic efficiency of the gas evolution reactions. Furthermore, the units for the evolved gases in the graph should be expressed as μmol/cm².

Reply: Many thanks for your valuable suggestions. The theoretical generation rates of O₂ and H₂ have been calculated based on the photocurrent densities. The Faradaic efficiency of the gas evolution reactions have been calculated.

The units for the evolved gases in the graph have been revised as μmol/cm². We attached the figure below for your information.

Fig. 4h Plots of the theoretical charge number obtained from the J–t curves collected at 1.23V vs. RHE and actual quantities of H₂ and O₂ evolution of a BVO-ΔO_v/FeOOH photoanode. Exposed area of the photoanode: 0.126 cm².

Changes to the revised manuscript are shown below.

Main Manuscript (Results):

Page 22: As shown in Fig. 4h, the produced H₂ and O₂ gases are 589.7 and 257.4 μmol cm⁻² after 5 h, respectively, indicating a stoichiometric ratio of around 2:1 for water splitting with an average faradaic efficiency of 97.6% (Supplementary Equation 15).

5. The authors must present the data on photoelectrochemical (PEC) evaluation more clearly. It remains unclear which irradiation source was utilized in this study. Was all data obtained using a Xe lamp with a 1.5 AM filter, or was a solar simulator employed? The reviewer suggests that the authors provide additional details regarding the irradiation source and elaborate on the calibration procedure for this source. Ideally, the authors should present the calibrated spectrum of the source used in this investigation and compare it with the AM 1.5G solar irradiance spectrum at various wavelengths.

Reply: Thanks a lot for your excellent suggestions. All data in our manuscript was obtained using a Xe lamp with an AM 1.5G filter. The light intensity was carefully calibrated to be ~100 mW cm⁻² by a light power meter. The calibrated spectrum of our light source is shown in Supplementary Fig. 1.

Supplementary Fig. 1. Spectrum of the Xe lamp light equipped with an AM 1.5G filter. A spectral irradiance of the Xe lamp light equipped with an AM 1.5G filter in the range of 300-800 nm was carefully calibrated to well match that of the standard AM 1.5G spectrum.

Changes to the revised manuscript are shown below.

Main Manuscript (Results):

Page 8: The Xe light equipped with an AM 1.5G filter was carefully calibrated to well match the standard AM 1.5G spectrum in the range of 300-800 nm (Supplementary Fig. 1).

6. In Figure 4g, the authors report results from stability tests conducted at a potential of 0.6 V_{RHE} under simulated solar conditions. It appears that the temperature of the electrolyte was not maintained at a constant level during these stability tests. The observed increase in photocurrent from the beginning of the experiment may be attributed to thermal heating of the electrolyte due to solar infrared radiation. The authors are advised to address this potential concern.

Reply: We appreciate your insightful comments. In our experiments, we monitored the temperature and found that the fluctuation of temperature will definitely affect the photocurrent density. As shown in Supplementary Fig. 26, the photocurrent density fluctuates with the same trend of the temperature. To avoid the effect to temperature change during the daytime and night in our lab, we have designed a reactor with constant temperature circulating water to keep the temperature of the electrolyte at 25 °C during the stability test. However, small fluctuations of the photocurrent densities can still be observed, which is due to the slight fluctuation of the output voltage and current of the Xe lamp during long-term testing.

To avoid any misunderstanding, we have added several sentences in the main text for explanation of this phenomenon, as highlighted in Page 21-22 in the revised manuscript.

Changes to the revised manuscript are shown below.

Main Manuscript (Results):

Page 21: We found that the fluctuation of the temperature in the electrolyte affects the stability curve (Supplementary Fig. 26). To avoid the effect of temperature on the stability measurement, we designed a reactor with constant temperature circulating water to keep the temperature of the electrolyte at 25 °C during the stability test.

During the stability test, slight fluctuations of the photocurrent densities can still be

observed, which is due to the fluctuation of the output power of the Xe lamp during the long-term operation.

Supplementary Fig. 26. Effect of the temperature on the photocurrent density. J-t curve of a BVO- ΔO_v /FeOOH photoanode in a 1 M borate buffer electrolyte (pH 9.5) under AM 1.5G illumination and the change of temperature in the electrolyte.

7. Figure 5e: The authors report STH value derived from intersection points and photocurrent density using a specific equation. The STH efficiency should be measured with respect to the V_{RHE} scale but should also account for measurements in a two-electrode system relevant to the V_{CE} scale. To accurately determine the real STH value, the authors should calculate the STH efficiency based on gas evolution using Faraday's law of electrolysis. This value will represent the measured STH efficiency of the tandem device.

Reply: Many thanks for your excellent suggestions. We have measured the J-V curve of BiVO₄ respect to the V_{CE} scale, which is similarly to the one respect to the V_{RHE} scale. Figure 5e has been updated by the curves in the V_{CE} scale. Gas evolution performances of wireless artificial leaves with dimensions up to 3 cm × 3 cm were measured in a sealed reactor connecting to a GC (Supplementary Fig 40).

Additional discussion has been added in the main text, as highlighted in Page 25 in the revised manuscript.

Changes to the revised manuscript are shown below.

Main Manuscript (Results):

Page 26-27: Gas evolution performances of wireless artificial leaves with dimensions up to 3 cm × 3 cm were measured in a sealed reactor connecting to a GC (Supplementary Fig. 40).

The gas production performance 3 wireless artificial leaves were measured, and the average H₂ and O₂ evolution performances with error bars are shown in Supplementary Fig. 42.

Based on the gas evolution performances (Supplementary Figs. 42, 43) and Supplementary Equation 19, the STH efficiency of the artificial leaf with an exposed area of 0.126 cm² can be calculated as 8.4%. It can be observed that the STH efficiencies calculated from the production of hydrogen are slightly lower than those calculated from the operating photocurrent densities, which is because the Faradaic efficiency for PEC water splitting is not 100%.

8. Page 20: In this section, the authors discuss the fabrication of a large-scale tandem device with a surface area of approximately 441 cm² (Figure 5d) and demonstrate its operational principle under natural sunlight. However, it would be preferred if the authors could provide the actual STH efficiency of the larger tandem device when operating under natural sunlight. Furthermore, should the present system be scaled up to a panel size of 1 m², the fabricated device is expected to produce a mixture of hydrogen and oxygen gases. This raises concerns about the effective separation of hydrogen from the proposed system. It would be advantageous for the authors to address this issue, detailing the methodologies or technologies that could be implemented to facilitate the efficient separation of hydrogen.

Reply: Thanks a lot for your constructive suggestions. The large-scale tandem device with a surface area of 441 cm² was put in a sealed reactor, and placed outdoors under the illumination of natural sunlight. The produced gases were collected by a syringe every 30 min and injected into a gas chromatography. The power density of the natural sunlight was measured by a light power meter. The actual STH efficiency was calculated to be 2.7%.

It is true that the standalone artificial leaf would inevitably lead to the mixture of

hydrogen and oxygen gases, which requires another procedure to separate the hydrogen and oxygen gases. For future possible scale-up applications, we can address this issue by connecting the Pt electrode with a wire as the counter electrode. In this case, we can use a sealed H-type reactor where the PEC-PV device is placed in Chamber 1 and the Pt electrode is placed in Chamber 2. Chamber 1 and Chamber 2 are separated by a Nafion membrane. Under light illumination, oxygen gas would be collected in Chamber 1, while hydrogen gas would be collected in Chamber 2. In this design, we can collect the separated hydrogen and oxygen gases, respectively.

Additional discussion has been added in the main text to address this concern, as highlighted in Page 27 in the revised manuscript.

Changes to the revised manuscript are shown below.

Main Manuscript (Results):

Page 26-27: Gas evolution performances of wireless artificial leaves with dimensions up to 3 cm × 3 cm were measured in a sealed reactor connecting to a GC (Supplementary Fig. 40). Larger area artificial leaves with dimensions of 6 cm × 6 cm (exposed area: 25 cm²), 9 cm × 9 cm (exposed area: 56.25 cm²), 12 cm × 12 cm (exposed area: 100 cm²) and 21 cm × 21 cm were sealed in home-made quartz reactors with dimensions of 15 cm × 15 cm × 10 cm and 30 cm × 30 cm × 10 cm (Supplementary Fig. 41), and the produced gas samples were taken out with a syringe, and injected in GC every hour for testing. The gas production performance 3 wireless artificial leaves were measured, and the average H₂ and O₂ evolution performances with error bars are shown in Supplementary Fig. 42. It can be observed that the performance of the wireless artificial leaf is similar to that of the wired artificial leaf (Fig. 5h). Based on the gas evolution performances (Supplementary Figs. 42, 43) and Supplementary Equation 19, the STH efficiencies of the artificial leaves with exposed areas of 0.126 cm², 1 cm², 6.25 cm², 25 cm², 56.25 cm², 100 cm² and 306.25 cm² can be calculated as 8.4%, 8.2%, 7.2%, 6.3%, 5.0%, 4.2%, and 2.7% (Supplementary Fig. 44).

For standalone artificial leaves, the production of a mixture of H₂ and O₂ gases may cause safety issues and requires the subsequent separation of H₂ and O₂. This issue can be addressed by assembling the wired artificial leaf in an H-type reactor with two

chambers (denoted as Chamber 1 and Chamber 2) separated by a Nafion membrane. Specifically, the encapsulated BVO- $\Delta\text{O}_v/\text{FeOOH-PV}$ photoanode is placed in Chamber 1, while a Pt counter electrode is placed in Chamber 2. The BVO- $\Delta\text{O}_v/\text{FeOOH-PV}$ photoanode and the Pt counter electrode is connected through the external circuit. Under light illumination, O_2 can be collected in Chamber 1, whereas H_2 can be collected in Chamber 2. In this design, we can collect the separated H_2 and O_2 gases, respectively.

Supplementary Fig. 42. Gas evolution performance for a wireless artificial leaf. Average H_2 and O_2 evolution performances of 3 artificial leaves with dimensions $3\text{ cm} \times 3\text{ cm}$ (exposed area 6.25 cm^2). Error bars are included.

Supplementary Fig. 43. Gas evolution performance for wireless artificial leaves with different sizes. **(a)** $1\text{ cm} \times 1\text{ cm}$ (control exposed area 0.126 cm^2), **(b)** $1\text{ cm} \times 1\text{ cm}$ (exposed area: 1 cm^2), **(c)** $6\text{ cm} \times 6\text{ cm}$ (exposed area: 25 cm^2), **(d)** $9\text{ cm} \times 9\text{ cm}$ (exposed area: 56.25 cm^2), **(e)** $12\text{ cm} \times 12\text{ cm}$ (exposed area: 100 cm^2) and **(f)** $21\text{ cm} \times 21\text{ cm}$ (exposed area: 306.25 cm^2).

Supplementary Fig. 44. STH efficiencies for wireless artificial leaves with different sizes. STH efficiencies of wireless artificial leaves with different sizes calculated based on their hydrogen evolution performances: 1 cm × 1 cm (control exposed area 0.126 cm²), 1 cm × 1 cm (exposed area: 1 cm²), 3 cm × 3 cm (exposed area 6.25 cm²), 6 cm × 6 cm (exposed area: 25 cm²), 9 cm × 9 cm (exposed area: 56.25 cm²), 12 cm × 12 cm (exposed area: 100 cm²) and 21 cm × 21 cm (exposed area: 306.25 cm²).

9. Figure 5h: The data presented concerning the H₂/O₂ evolution from the BVO-PV tandem device raises several questions for the reviewer, particularly regarding the rationale for replacing the electrolyte every 10 hours during the testing procedure. If the device requires fresh electrolyte at this interval, it could present significant challenges for scaling up the system for practical applications. Additionally, it appears that the rates of H₂ and O₂ production decrease throughout the testing period. The authors are encouraged to address these concerns.

Reply: Thank you for pointing out this important issue for us. We are sorry that we carelessly making the inaccurate expressions in the manuscript that causes unnecessary confusion. Since our photoanode can continuously achieve PEC water splitting for over 500 h (Fig. 4g), and there is no any scavenger agent to be consumed in the electrolyte,

it is not necessary to replace the electrolyte every 10 h. We meant that we vacuumed the system every 10 hours to test the gas evolution, but we did not replace the electrolyte. We apologize for making the confusion.

Figure 5h in the previous version was measured using a wireless artificial leaf with dimensions of 3 cm × 3 cm (exposed area 6.25 cm²). To confirm the H₂ and O₂ production performance, we used a wired artificial leaf with dimensions of 3 cm × 3 cm (exposed area 6.25 cm²) to monitor the photocurrent densities and the gas evolution performance. As shown in Figure 5g, the photocurrent density retention rate is 92% after 50 h. The H₂ and O₂ production rates are almost linear in all cycles, and the Faradaic efficiency is around 97% (Figure 5h), indicating that almost all the photocurrents are used for water splitting.

In the case of a wireless artificial leaf, the H₂ and O₂ production rates exhibit some slight fluctuations during the testing period, which may due to the random measurement error for the test. However, the overall trends of H₂ and O₂ productions are linear increase. To reduce the random measurement error, we tested 3 wireless artificial leaves with dimensions of 3 cm × 3 cm (exposed area 6.25 cm²) and added the error bars for H₂ and O₂ productions. As shown in Supplementary Fig. 42, the H₂ and O₂ production rates are similar to those of the wired system (Figure 5h).

Changes to the revised manuscript are shown below.

Main Manuscript (Results):

Page 25: A wired BVO-ΔO_v/FeOOH-PV artificial leaf with dimensions of 3 cm × 3 cm was placed in a sealed reactor connecting to an electrochemical workstation to monitor the photocurrent densities (Supplementary Fig. 38). The BVO-ΔO_v/FeOOH-PV artificial leaf can achieve water splitting for 50 h with a photocurrent density retention rate of 92% (Fig. 5g).

Page 26: The system was vacuumed every 10 h to remove all gases for one cycle test. As shown in Fig. 5h, the generated H₂/O₂ show no noticeable decrease, with production rates of 1117/551 μmol cm⁻² in 10 h of consecutive illumination, with a ratio of round 2:1.

The gas production performance of 3 wireless artificial leaves were measured, and the average H_2 and O_2 evolution performances with error bars are shown in Supplementary Fig. 42. It can be observed that the performance of the wireless artificial leaf is similar to that of the wired artificial leaf (Fig. 5h).

Fig.5g Photocurrent density-time curve of a wired BVO- ΔO_v /FeOOH-PV artificial leaf with dimensions of $3\ cm \times 3\ cm$ (exposed area: $6.25\ cm^2$) under AM 1.5 G irradiation.

h The corresponding H_2 and O_2 evolution of a of a wired BVO- ΔO_v /FeOOH-PV artificial leaf with dimensions of $3\ cm \times 3\ cm$ (exposed area: $6.25\ cm^2$) under AM 1.5 G irradiation.

Supplementary Fig. 42. Gas evolution performance for a wireless artificial leaf. Average H_2 and O_2 evolution performances of 3 artificial leaves with dimensions $3\ cm \times 3\ cm$ (exposed area $6.25\ cm^2$). Error bars are included.

10. In Figures 5g and h, the unit for photocurrent density should be specified in mA/cm², and the unit for evolved gases must be indicated as μmol/cm².

Reply: Many thanks for your suggestions. The unit for photocurrent density in Figure 5g has been changed to mA/cm², and the unit for evolved gases in Figure 5h has been changed to μmol/cm². We attached the figure below for your information.

Fig. 5g Photocurrent density-time curve of a wired BVO-ΔO_v/FeOOH-PV artificial leaf with dimensions of 3 cm × 3 cm (exposed area: 6.25 cm²) under AM 1.5 G irradiation. **h** The corresponding H₂ and O₂ evolution of a of a wired BVO-ΔO_v/FeOOH-PV artificial leaf with dimensions of 3 cm × 3 cm (exposed area: 6.25 cm²) under AM 1.5 G irradiation.

Response to Reviewers' Comments

Reviewer #1 (Remarks to the Author):

Authors arranged the revision part exceptionally well based on my previous comments and now manuscript seems suitable for publication in the current form. With following minor suggestion in the main manuscript, it is advisable to add one sentence and citing relevant proper references for origin of gridline design ideas in photoelectrodes.

Joule, 7, 884–919, 2023;

Chem. Soc. Rev., 53, 2388-2434, 2024;

Int. J. Hydrogen Energy 36, 5262-5270, 2011;

J. Power Sources, 398, 224-232, 2018;

J. Power Sources, 454, 227890, 2020;

Sustainable Energy Fuels, 8, 3726-3739, 2024;

Sustainable Energy Fuels, 3, 2366-2379, 2019

Reply: Thank you very much for your affirmation of our revision, and your excellent suggestions to further improve the quality of our manuscript. Additional discussion has been added in the main text, as highlighted in Page 27 in the revised manuscript. The recommended papers have been cited as Refs. 48, 51, 52, 53, 55, 56, and 58.

Changes to the revised manuscript are shown below.

Main Manuscript (Results):

Page 27: The ohmic loss issue can be addressed by embedding metal gridlines (e.g., Ag, Ni) on FTO substrates^{48,55}. In our work, the large-area photoanode was assembled by sub-photoanodes (dimensions: 3 cm × 3 cm) connected by conductive gridlines, which is also effective to reduce the ohmic losses within the film⁵⁶.

References:

48 Ahmet, I. Y. *et al.* Demonstration of a 50 cm² BiVO₄ tandem photoelectrochemical-photovoltaic water splitting device. *Sustainable Energy Fuels* **3**, 2366-2379 (2019).

51 Vilanova, A., Lopes, T. & Mendes, A. Large-area photoelectrochemical water splitting using a multi-photoelectrode approach. *J. Power Sources* **398**, 224-232 (2018).

52 Vilanova, A. *et al.* Solar water splitting under natural concentrated sunlight using

a 200 cm² photoelectrochemical-photovoltaic device. *J. Power Sources* **454**, 227890 (2020).

53 Maragno, A. R. A. *et al.* Thermally integrated photoelectrochemical devices with perovskite/silicon tandem solar cells: a modular approach for scalable direct water splitting. *Sustainable Energy Fuels* **8**, 3726-3739 (2024).

55 Lee, W. J., Shinde, P. S., Go, G. H. & Ramasamy, E. Ag grid induced photocurrent enhancement in WO₃ photoanodes and their scale-up performance toward photoelectrochemical H₂ generation. *Int. J. Hydrogen Energy* **36**, 5262-5270 (2011).

56 Hansora, D., Cherian, D., Mehrotra, R., Jang, J.-W. & Lee, J. S. Fully inkjet-printed large-scale photoelectrodes. *Joule* **7**, 884-919 (2023).

58 Vilanova, A., Dias, P., Lopes, T. & Mendes, A. The route for commercial photoelectrochemical water splitting: a review of large-area devices and key upscaling challenges. *Chem. Soc. Rev.* **53**, 2388-2434 (2024).

Reviewer #2 (Remarks to the Author):

Since the authors have extensively addressed all the comments and issues raised by the referees, I recommend publication of this manuscript in Nature Communications.

Reply: We appreciate your constructive suggestions to improve the quality of our manuscript.

Reviewer #3 (Remarks to the Author):

I have completed the second revision of the manuscript titled “A Standalone Bismuth Vanadate-Silicon Artificial Leaf Achieving 8.4% Efficiency for Zero-Emission Hydrogen Production.” Upon review, I note that the authors have addressed the referee's comments and suggestions with due diligence. I recommend the manuscript for publication in Nature Communications, pending the incorporation of the minor revisions detailed below.

Reply: We are grateful for your guidance and great suggestions to improve the quality of our work. We have carefully considered all your suggestions, and have revised the manuscript accordingly. Please see our revision and point-by-point responses shown below.

1. Figure 4h: The current presentation of this figure appears overly complex. I suggest the authors consider removing the data corresponding to photocurrent measurements at 1.23 V versus RHE from this figure. Instead, this data could be effectively integrated into Figure 4g. Moreover, if the authors have conducted stability measurements of the photoanode at 1.23 V versus RHE over an extended duration, it would be beneficial to include this data in Figure 4g as well.

Reply: Many thanks for your valuable suggestions. We have removed the photocurrent curve at 1.23 V vs. RHE from Figure 4h. In our 1st revision, the stability measurements of the photoanode at 1.23 V versus RHE were shown in Supplementary Fig. 30. We have added this data in Figure 4g. We attached the revised figure below for your information.

Fig. 4. Material characterizations and photoelectrochemical performances. **a** SEM image of BVO- ΔO_v /FeOOH. **b** Enlarged SEM image of BVO- ΔO_v /FeOOH. **c** LSV curves in the dark, **(d)** Charge transfer efficiency (η_{trans}) curves, **(e)** LSV curves in the light, **(f)** ABPE curves of the BVO- ΔO_v and BVO- ΔO_v /FeOOH photoanodes in a 1 M borate buffer electrolyte (pH 9.5) under AM 1.5G illumination. **g** J-t curve of the BVO-

$\Delta\text{O}_v/\text{FeOOH}$ photoanode at 0.6 and 1.23 V vs. RHE under AM 1.5G illumination in a 1 M borate buffer electrolyte (pH 9.5). **h** Plots of the theoretical charge number obtained from the J–t curves collected at 1.23V vs. RHE and actual quantities of H₂ and O₂ evolution of a BVO- $\Delta\text{O}_v/\text{FeOOH}$ photoanode. Exposed area of the photoanode: 0.126 cm².

2. Methods Section: The authors are advised to specify the model of the light power meter employed to calibrate the Xe-lamp for clarity and reproducibility.

Reply: Thank you for your excellent suggestions. The model of the light power meter for calibrating the Xe-lamp has been added in the Methods Section, as highlighted in Page 36 in the revised manuscript.

Changes to the revised manuscript are shown below.

Main Manuscript (Methods):

Page 36: The light source was a Xe 300 W lamp (CEL-S300, CEAULIGHT) with an AM 1.5 G filter, and the light intensity at the WE was calibrated to 100 mW cm⁻² using an optical power meter (CEL-NP2000-2A, CEAULIGHT). The spectrum of the light source was confirmed to match the standard AM 1.5G spectrum using a MAX2000-Pro spectroradiometer (Shanghai Wyoptics Technology Co., Ltd).

3. Regarding Previous Inquiry 8 (Oxygen and Hydrogen Separation): The authors propose the use of an H-type reactor with two chambers separated by a Nafion membrane for gas separation. However, it is important to note that the conductivity of the Nafion membrane is temperature-dependent, with proton exchange membrane devices typically operating near 80 °C. I encourage the authors to provide a more detailed discussion regarding the future development of large-scale systems that can efficiently separate hydrogen, including considerations for optimizing the operating conditions.

Reply: Thanks a lot for your insightful suggestions. Yes, the efficient separation of hydrogen and oxygen at different operating conditions are very important for scale-up applications. We have carefully researched the literature and found some useful information. Detailed discussion has been added in the main text, as highlighted in Page

28 in the revised manuscript.

Changes to the revised manuscript are shown below.

Main Manuscript (Results):

Page 28: It should be mentioned that the protonic conductivity of a Nafion membrane increases significantly with temperature ($< 80\text{ }^{\circ}\text{C}$)⁵⁷. Since solar water splitting is generally operated at room temperature, the performance of this configuration may be limited by the resistance of the Nafion membrane⁵⁸. On the other hand, membrane-free devices with liquid- or solid-state redox mediators can generate H₂ and O₂ gases in separated cells during water splitting^{59,60}, which can eliminate the resistance caused by the incorporation of membranes. However, owing to the increase distance between the two electrodes, the ohmic losses of the system also increase. More works are required to confirm the reliability of membrane-free devices for scale-up applications.

References:

57 Zhang, L., Chae, S.-R., Hendren, Z., Park, J.-S. & Wiesner, M. R. Recent advances in proton exchange membranes for fuel cell applications. *Chem. Eng. J.* **204-206**, 87-97 (2012).

58 Vilanova, A., Dias, P., Lopes, T. & Mendes, A. The route for commercial photoelectrochemical water splitting: a review of large-area devices and key upscaling challenges. *Chem. Soc. Rev.* **53**, 2388-2434 (2024).

59 Slobodkin, I., Davydova, E., Sananis, M., Breytus, A. & Rothschild, A. Electrochemical and chemical cycle for high-efficiency decoupled water splitting in a near-neutral electrolyte. *Nat. Mater.* **23**, 398-405 (2024).

60 Dotan, H. *et al.* Decoupled hydrogen and oxygen evolution by a two-step electrochemical–chemical cycle for efficient overall water splitting. *Nat. Energy* **4**, 786-795 (2019).

4. Supplementary Figure 14: This figure illustrates the solar-to-hydrogen (STH) efficiencies of wireless artificial leaves of varying device sizes. It is evident from the data that the STH efficiency is size-dependent, with a noted decrease to 2.7% when the device size reaches 306 cm². The authors are encouraged to provide a comprehensive explanation of the primary factors affecting STH efficiency as device size increases and to discuss potential strategies to mitigate this observed decline.

Reply: Thank you very much for your constructive suggestions. The explanation of the size-dependent STH efficiency is important for our study. According to our understanding and a comprehensive literature review, the detailed discussion has been added in the main text, as highlighted in Page 27-28 in the revised manuscript.

Changes to the revised manuscript are shown below.

Main Manuscript (Results):

Page 27-28: It can be observed that the STH efficiency of the device decreases with the increase of the size (Supplementary Fig. 43), which is caused by many factors such as ohmic losses in the FTO substrate, the electrolyte conductivity, and mass-transport (H^+/OH^- ions) limitations⁴⁸. The ohmic loss issue can be addressed by embedding metal gridlines (e.g., Ag, Ni) on FTO substrates^{48,55}. In our work, the large-area photoanode was assembled by sub-photoanodes (dimensions: 3 cm × 3 cm) connected by conductive gridlines, which is also effective to reduce the ohmic losses within the film⁵⁶. The electrolyte conductivity can be enhanced by increasing the buffer concentration in the electrolyte⁴⁸. The formation of proper through-holes in the FTO substrates would provide more paths for the transport of the H^+/OH^- ions during PEC water splitting, which may be helpful to mitigate the mass-transport issues. However, the loading mass of the photoanode materials is also reduced, which would inevitably decrease the light absorption efficiency. In addition, high concentration acidic or alkaline electrolytes are beneficial for the transport of the H^+/OH^- ions. However, it is a big challenge to achieve long-term stability for the device in such a harsh environment.

References:

- 48 Ahmet, I. Y. *et al.* Demonstration of a 50 cm² BiVO₄ tandem photoelectrochemical-photovoltaic water splitting device. *Sustainable Energy Fuels* **3**, 2366-2379 (2019).
- 55 Lee, W. J., Shinde, P. S., Go, G. H. & Ramasamy, E. Ag grid induced photocurrent enhancement in WO₃ photoanodes and their scale-up performance toward photoelectrochemical H₂ generation. *Int. J. Hydrogen Energy* **36**, 5262-5270 (2011).
- 56 Hansora, D., Cherian, D., Mehrotra, R., Jang, J.-W. & Lee, J. S. Fully inkjet-printed large-scale photoelectrodes. *Joule* **7**, 884-919 (2023).

Response to Reviewers' Comments

Reviewer #3 (Remarks to the Author):

I have read the paper “A standalone bismuth vanadate-silicon artificial leaf achieving 8.4% efficiency for zero-emission hydrogen production” after final revision and I can clearly say that the authors revised the manuscript following all the referees' comments and suggestions. The current version of the manuscript can be recommended for publication in Nature Communications.

Reply: Thank you very much for all your valuable suggestions to help improving the quality of our work, and many thanks for your approval on our revision.